# TRUE SELF-SUPERVISED NOVEL VIEW SYNTHESIS IS TRANSFERABLE

**Thomas W. Mitchel**[*]
Adobe, PlayStation
thomas.w.mitchel@gmail.com

**Hyunwoo Ryu**[*]
MIT CSAIL, PlayStation
hwryu@mit.edu

**Vincent Sitzmann**
MIT CSAIL
sitzmann@mit.edu

## ABSTRACT

In this paper, we identify that the key criterion for determining whether a model is truly capable of novel view synthesis (NVS) is *transferability*: Whether any pose representation extracted from one video sequence can be used to re-render the same camera trajectory in another. We analyze prior work on self-supervised NVS and find that their predicted poses do *not* transfer: The same set of poses lead to *different* camera trajectories in different 3D scenes. Here, we present *XFactor*, the first geometry-free self-supervised model capable of *true* NVS. XFactor combines pair-wise pose estimation with a simple augmentation scheme of the inputs and outputs that jointly enables disentangling camera pose from scene content and facilitates geometric reasoning. Remarkably, we show that XFactor achieves transferability with unconstrained latent pose variables, without any 3D inductive biases or concepts from multi-view geometry — such as an explicit parameterization of poses as elements of $SE(3)$. We introduce a new metric to quantify transferability, and through large-scale experiments, we demonstrate that XFactor significantly outperforms prior pose-free NVS transformers, and show that latent poses are highly correlated with real-world poses through probing experiments. Project Page: https://www.mitchel.computer/xfactor/

## 1 INTRODUCTION

In recent years, novel view synthesis (NVS) has emerged as a canonical 3D computer vision problem. Methods today are built on the rich discipline of multi-view geometry, which has given rise to structure-from-motion models that can preprocess large datasets of multi-view images to compute corresponding $SE(3)$ camera poses. Given a dataset of multi-view images and their camera poses, state-of-the-art methods allow a user to specify a camera pose as an $SE(3)$ transform and render the corresponding view near photorealistically. However, the bitter lesson (Sutton, 2019) teaches us to be skeptical of any inductive bias in learning systems. In this paper, we ask: Can we formulate NVS *without* reliance on multi-view geometry, tackling it as a *pure* machine learning problem?

To answer this question, we must first ask what novel view synthesis is without relying on the vocabulary of multi-view geometry. To this end, we identify *transferability* as the key property of any novel view synthesis model: the ability to use a set of camera poses extracted from one sequence to render the *same* camera trajectory in any other scene. As a corollary, the key requirement of any valid representation of camera poses is *not* that they can be identified with an $SE(3)$ representation, but that they render the same camera trajectory across scenes.

Equipped with this insight, we tackle self-supervised novel view synthesis as a pure machine learning problem. We find that existing methods (Jiang et al., 2025; Sajjadi et al., 2023) *do not* infer transferable camera poses, and are instead prone to interpolating context frames. This is *not* true novel view synthesis, as it does not allow the user to define which view they want to render in an arbitrary scene. Here, we present *XFactor*, the first self-supervised model capable of *true* NVS. XFactor combines pair-wise pose estimation with a simple augmentation scheme of the inputs and outputs that jointly enables disentangling camera pose from scene content and facilitates geometric reasoning. This is motivated by two key insights: 1) preventing the model from interpolating by

---

[*]Equally contributing authors.

bootstrapping from a two-view NVS model that extrapolates by design, 2) reifying transferability into a training objective compatible with real-world video by augmenting sequences of frames in a manner that minimizes pixel content overlap while preserving camera motion, such as applying two inverse masks to the same sequence. XFactor achieves transferability with unconstrained latent pose variables, without any 3D inductive biases or concepts from multi-view geometry — such as an explicit parameterization of poses as elements of SE(3). We then fine-tune the two-view XFactor model into a multi-view model that we show enables transferable, high-quality NVS: Given a sequence of frames and choosing one as the reference, we can generate a latent trajectory by using the encoder to estimate the pose between the reference and each frame; Then, using any other video sequence as context, our renderer will reproduce that same camera trajectory in the new scene.

Through extensive experiments, we show that our method is the first fully geometry-free[1] and self-supervised method that achieves true NVS across diverse, large-scale real-world datasets at both the scene and object level — including RE10K (Zhou et al., 2018), DL3DV (Ling et al., 2024), MVImgNet (Yu et al., 2023), and CO3Dv2 (Reizenstein et al., 2021). In particular, we introduce a metric for quantifying the degree to which novel views adhere to reference poses, and demonstrate that XFactor dramatically outperforms prior methods RayZer (Jiang et al., 2025) and RUST (Sajjadi et al., 2023). In a series of ablations, we analyze what design decisions matter to solve transferable novel view synthesis, and demonstrate that, counter-intuitively, forcing the model to parameterize camera poses as SE(3) is harmful and rather, what matters is a careful design of inputs and outputs to the pose estimator and renderer. In summary, we make the following key contributions:

1. We introduce *transferability* as the key criterion for determining whether a self-supervised model is capable of *true* NVS and introduce the True Pose Similarity metric to quantify it.

2. We identify that prior multi-view self-supervised NVS models interpolate context frames instead of reasoning about viewpoints. We address this by boot-strapping multi-view NVS on top of a two-frame model which, by design, always extrapolates.

3. We propose a novel self-supervised NVS training objective which explicitly promotes transferability, and introduce a representation learning-inspired augmentation strategy for training with real-world video.

4. Derived from these insights, we present XFactor which to our knowledge is the first fully self-supervised NVS model to achieve transferability and thus perform *true* NVS.

5. We empirically demonstrate the merits of our formulation in comprehensive large-scale experiments and ablations.

## 2 RELATED WORK

We review prior work on novel view synthesis *with* and *without* known camera poses.

**Oracle, Semi-Oracle, and Geometric Methods.** Using camera poses obtained from an external pose oracle, such as COLMAP (Schönberger & Frahm, 2016a), neural networks can be trained to predict 3D neural scene representations (Yu et al., 2021; Charatan et al., 2023) or novel views directly (Jin et al., 2024; Sajjadi et al., 2021; Sitzmann et al., 2021). We refer to these methods as "Oracle Methods". Recent work has attempted to reduce the reliance on poses at training time to tap into larger datasets. These methods work by training a pose prediction module jointly with the novel view synthesis module. However, existing methods nevertheless rely on some form of external oracle, such as pre-trained optical flow or correspondence methods (Smith et al., 2023; Chen & Lee, 2023; Wang et al., 2023), pre-trained depth estimators (Fu et al., 2023; Brachmann et al., 2024), or initialization with weights that were pre-trained on a supervised structure-from-motion task (Huang & Mikolajczyk, 2025b). Some prior work achieves impressive camera pose estimation *without* relying on pre-trained oracles (Kang et al., 2025; Yin & Shi, 2018; Zhou et al., 2017), enabled by strong expert-crafted geometric inductive biases such as warping, correspondence matching, and depth prediction. The goal of our paper is to develop a first-principles approach to novel view synthesis that does not rely on *any* form of conventional multi-view geometry.

---

[1]We use the term 'geometry-free' to denote that the model is free of heuristic 3D representations such as Gaussian splats, volumetric rendering or Plücker embeddings.

**Unsupervised Geometry-Free Latent Pose Methods.** While several works have addressed unposed novel view synthesis via explicit 3D modeling (Bian et al., 2023; Levy et al., 2024; Ye et al., 2024; Wang et al., 2025c; Kang et al., 2025; Huang & Mikolajczyk, 2025a), only a small number of methods have attempted to solve the unposed novel view synthesis problem without relying on external 3D oracles and using as little 3D inductive bias as possible, tackling NVS as a pure deep learning problem. In this case, a pose estimation module predicts some form of camera poses that are used as conditioning inputs to a geometry-free renderer transformer. A key challenge is to prevent the pose estimator from communicating information about the target frames to the renderer. RayZer (Jiang et al., 2025) and Less3Depend (Wang et al., 2025a) attempt to accomplish this by parameterizing latent poses as rigid-body SE(3) transforms. While renders are high quality, we show that this approach has a significant limitation: The same set of predicted poses renders *different* camera trajectories in different 3D scenes, i.e., camera poses do not *transfer* between scenes. As we will see, this is the effect of the renderer performing *interpolation* of context frames rather than true NVS. Closest to our work is RUST (Sajjadi et al., 2023), which attempts a fully geometry-free approach to novel view synthesis — its promising results were an inspiration for the present method. RUST attempts to prevent cheating via an information bottleneck: the pose estimator receives only *part* of the target view. However, RUST does not solve the transferability problem — our proposed method is the first method to achieve geometry-free true NVS. Finally, we note that our approach shares similarities with recent work on latent action models (Zhang et al., 2025; Gao et al., 2025; Bruce et al., 2024; Schmidt & Jiang, 2024). While these models seek to extract transition latents describing a variety of ego-centric actions from adjacent video frames, we instead focus on the specific problem of recovering transferable camera pose representations.

## 3  METHOD

### 3.1  NOVEL VIEW SYNTHESIS AS LATENT VARIABLE MODELING

To isolate the key properties of NVS, we first formulate it as a latent variable model. Given a sequence of images $\mathcal{I} = \{I_1, I_2, \ldots, I_n\}$ of a static scene, existing NVS methods typically begin by partitioning them into two disjoint subsets of context images $\mathcal{I}_C$ and target images $\mathcal{I}_T$ with $\mathcal{I}_C \cup \mathcal{I}_T = \mathcal{I}$. These methods can generally be decomposed into three core components: a pose encoder POSEENC, a scene encoder SCENEENC, and renderer RENDER. Given a choice of reference view $I_R \in \mathcal{I}_C$ relative to which poses will be expressed, the pose encoder maps the context and target images to sets of latent pose representations; The scene encoder converts the context images and corresponding latent poses to a latent scene representation:

$$(\mathcal{I}_C, \mathcal{I}_T) \xmapsto{\text{POSEENC}} (\mathcal{Z}_C, \mathcal{Z}_T) \qquad \text{and} \qquad (\mathcal{I}_C, \mathcal{Z}_C) \xmapsto{\text{SCENEENC}} \mathcal{S}. \tag{1}$$

In the prevailing formulation consistent across both supervised (Jin et al., 2024) and unsupervised settings (Jiang et al., 2025; Sajjadi et al., 2023), the role of the rendering decoder is to synthesize a prediction of the target images from the target poses and latent scene representation

$$(\mathcal{S}, \mathcal{Z}_T) \xmapsto{\text{RENDER}} \widetilde{\mathcal{I}}_T, \tag{2}$$

and the model is trained to minimize what we call the autoencoding objective

$$L \equiv d_I\big(\mathcal{I}_T, \text{RENDER}[\mathcal{S}, \mathcal{Z}_T]\big), \tag{3}$$

subject to an image distance metric $d_I$. Satisfying this objective requires the model only to have the ability to render target frames using scene and pose representations *from the same sequence*.

An important auxiliary tool in this framework is the ORACLE, which is simply an algorithm that ingests a sequence of frames and spits out the ground-truth camera poses as elements of SE(3):

$$\{I_1, I_2, \ldots, I_n\} \xmapsto{\text{ORACLE}} \{g_1, g_2, \ldots, g_n\} \in \text{SE}(3)^n. \tag{4}$$

A canonical choice is ORACLE ≡ COLMAP (Schönberger & Frahm, 2016a). However, in this paper, we instead choose ORACLE ≡ VGGT (Wang et al., 2025b) due to its robustness and ease of use.

State-of-the-art feed-forward oracle NVS models including LVSM (Jin et al., 2024) and pixel-Splat (Charatan et al., 2023) simply define POSEENC ≡ ORACLE and seek to learn only SCENEENC

and RENDER. Similarly, single-scene oracle models such as those based on NeRF or Gaussian Splatting seek to directly optimize the scene representation $\mathcal{S}^B$ in the form of the weights of the rendering MLP or as the Gaussians themselves. In contrast, self-supervised NVS models RayZer (Jiang et al., 2025) and RUST (Sajjadi et al., 2023) aim to learn all three modules end-to-end without reliance on an ORACLE. However, we show empirically (Sec. 4.1) that the prevailing framework for NVS described in Equations (1–3) is in fact ill-suited for the self-supervised setting as it does not consider the fundamental property making a model capable of true NVS: *transferability*.

## 3.2 TRUE NOVEL VIEW SYNTHESIS IS TRANSFERABLE

NVS is simply the ability to render a scene from a *user-controllable* viewpoint: It is critical that the same camera pose always results in the same viewpoint being rendered. If the model *cannot* do this, it is *not* a true NVS model, but rather, a frame interpolator. We propose that in the perspective of NVS as a latent-variable model (Sec. 3.1), controllability is equivalent to *transferability* and can be formalized as the property that *pose representations transfer between scenes*.

Let $\mathcal{I}^A = \mathcal{I}_C^A \cup \mathcal{I}_T^A$ and $\mathcal{I}^B = \mathcal{I}_C^B \cup \mathcal{I}_T^B$ be sequences whose target frames $\mathcal{I}_T^A$ and $\mathcal{I}_T^B$ share the same camera motion, *i.e.* ORACLE$[\mathcal{I}_A^T] = $ ORACLE$[\mathcal{I}_B^T]$. Then, we say that an arbitrary NVS model consisting of core components [POSEENC, SCENEENC, RENDER] produces **transferable pose representations** (or is a **true NVS model**) if the latent representations

$$\text{POSEENC}\left[\mathcal{I}_C^A, \mathcal{I}_T^A\right] = \left(\mathcal{Z}_C^A, \mathcal{Z}_T^A\right) \qquad \text{and} \qquad \text{SCENEENC}\left[\mathcal{I}_C^B, \mathcal{Z}_C^B\right] = \mathcal{S}^B \tag{5}$$

and renderer satisfy

$$\text{RENDER}\left[\mathcal{S}^B, \mathcal{Z}_T^A\right] \approx \mathcal{I}_T^B. \tag{6}$$

This criterion automatically satisfies the autoencoding objective in Equation (3) when $\mathcal{I}^A = \mathcal{I}^B$ and captures the essence of controllable NVS: the ability to apply camera trajectories from one scene to synthesize views of another scene. We note that both conventional feed-forward *and* single-scene oracle NVS models with POSEENC $\equiv$ ORACLE are *automatically* transferable (and thus are capable of true NVS) with the autoencoding objective, as for any scene representation $\mathcal{S}^B$

$$\text{RENDER}\left[\mathcal{S}^B, \text{ORACLE}\left[\mathcal{I}_T^A\right]\right] = \text{RENDER}\left[\mathcal{S}^B, \text{ORACLE}\left[\mathcal{I}_T^B\right]\right] \overset{(3)}{\approx} \mathcal{I}_T^B. \tag{7}$$

## 3.3 QUANTIFYING TRANSFERABILITY WITH TRUE POSE SIMILARITY (TPS)

We introduce a standardized metric to quantify the degree of transferability of latent pose representations which we call **True Pose Similarity** (**TPS**). Given an ORACLE and trajectory comparison metric $d_{\text{SE}(3)^n}$, such as Relative Rotation Accuracy (RRA), Relative Translation Accuracy (RTA), or Area Under Curve (AUC) which combines the two, we define the TPS between two sequences of frames $\mathcal{I}^A$ and $\mathcal{I}^B$ of equal length to be the value of the metric between the oracle poses from each sequence:

$$\text{TPS}\left(\mathcal{I}^A, \mathcal{I}^B\right) \equiv d_{\text{SE}(3)^n}\left(\text{ORACLE}\left[\mathcal{I}^A\right], \text{ORACLE}\left[\mathcal{I}^B\right]\right). \tag{8}$$

Context Scene
Target Frames — Novel Views
Model
ORACLE — ORACLE
Same? Transferable!
Different? Not Transferable!

To quantify the transferability of a [POSEENC, SCENEENC, RENDER] NVS model with TPS, we consider two arbitrary sequences $\mathcal{I}^A = \mathcal{I}_C^A \cup \mathcal{I}_T^A$ and $\mathcal{I}^B = \mathcal{I}_C^B \cup \mathcal{I}_T^B$. We then use the scene representation from the second sequence $\mathcal{S}^B$ and the target latent poses $\mathcal{Z}_T^A$ from the first sequence as in Equation (1) to render a new trajectory in the second sequence, leveraging TPS to measure whether their camera trajectories agree:

$$\text{TPS}\left(\mathcal{I}_T^A, \text{RENDER}\left[\mathcal{S}^B, \mathcal{Z}_T^A\right]\right). \tag{9}$$

We note that this quantity only measures one component of transferability — that the rendered viewpoints are geometrically consistent, and not also faithful to the context sequence. For instance, this metric, unlike the definition in Sec. 3.2, can be hacked by a model RENDER$[\mathcal{S}^B, \mathcal{Z}_T^A] \approx \mathcal{I}_T^A$, so it is necessary to pair it with a perceptual measure — either qualitative or quantitative — to verify faithfulness. We highlight that we only rely on ORACLE for benchmarking purposes; our proposed method does not rely on any external pre-trained or expert-crafted ORACLE for training.

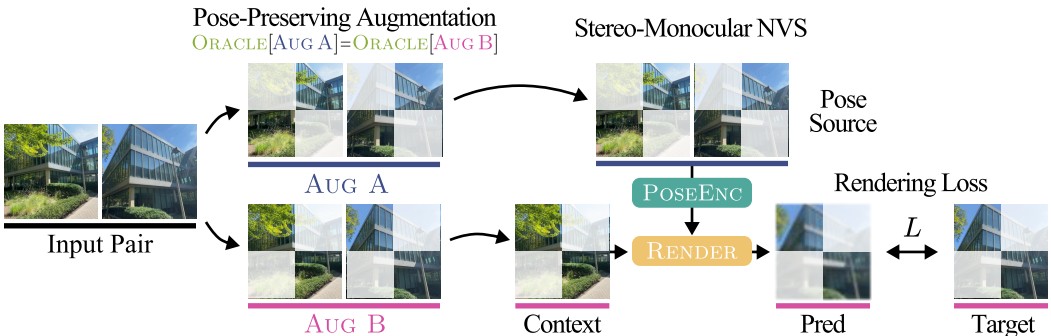

Figure 1: **XFactor** combines a [POSEENC, RENDER] *stereo-monocular model* with our proposed *transferability objective* to learn transferable camera pose latents. Given a pair of input images, we apply two augmentations that minimize pixel content overlap while preserving pose information, such as inverse masking. The stereo POSEENC extracts the relative pose latent from the first pair. Then, given the context image from the second pair and the first's target pose, the renderer is asked to reconstruct the second's target.

### 3.4 SOLVING TWO PRINCIPAL PROBLEMS: INTERPOLATION AND INFORMATION LEAKAGE

In the self-supervised setting there is no guarantee of transferability and we demonstrate empirically that existing models RayZer and RUST fail to produce transferable pose representations under the TPS metric (Sec. 4.1). In what follows we provide two key insights regarding why these models fail, and from them derive an approach for learning transferable pose latent representations.

**The Stereo-Monocular Model.** We first note that in both RayZer and RUST, their pose encoders and renderers have access to multiple context views. We find that training such a self-supervised multi-view model leads to a model that uses the latent "pose" to encode *how to interpolate context views to synthesize the target view*. Such a "pose" will *not* transfer to a different scene, as a different scene will feature *different* context views. This is hence *not* true NVS because it does not allow the user to define which view they want to render in an arbitrary scene.

To prevent the model from learning to interpolate and instead reason about poses, we propose to bootstrap a self-supervised multi-view NVS model off of a **stereo-monocular model** that must always *extrapolate*. Specifically, we consider the case where there is only a single context and target image, respectively — e.g. $\mathcal{I} = \{I_1, I_2\}$ with $\mathcal{I}_C = \{I_1\}$ and $\mathcal{I}_T = \{I_2\}$. Thus, the POSEENC becomes a *two-view stereo model*, SCENEENC *can be absorbed by* RENDER, and RENDER is *monocular*:

$$\text{POSEENC}\,[I_1,\,I_2] = Z_2 \qquad \text{and} \qquad \text{RENDER}\,[I_1,\,Z_2] = \widetilde{I_2}. \tag{10}$$

By providing RENDER with only a single image for reconstruction we eliminate the interpolation path and guide optimization toward learning transferable pose representations. We note that this approach shares similarities with CroCo (Weinzaepfel et al., 2023), a representation-learning method which leverages a monocular renderer to promote the learning of depth cues.

**The Transferability Objective.** While we show that the stereo-monocular model produces transferable pose representations (Sec. 4.3), it still allows for POSEENC to encode information about target pixels, rather than a purely geometric description of the relative pose. This again provides an easier "off ramp" for RENDER, which does not have to perform full NVS but can cheat by decoding pixel information smuggled into the target pose latent.

We propose to discourage the entanglement of pixel information by explicitly defining the training objective as transferability: Given two pairs of frames $\mathcal{I}^A = \{I_1^A,\,I_2^A\}$ and $\mathcal{I}^B = \{I_1^B,\,I_2^B\}$ which are known to share *the same relative camera pose* we ask that the relative pose latent extracted from the first sequence must be able to render the target image from the second,

$$L \equiv d_I\left(I_2^B, \text{RENDER}\left[I_1^B, \text{POSEENC}\left[I_1^A,\,I_2^A\right]\right]\right), \tag{11}$$

which we call the **transferability objective**. However, despite the obvious benefits of imposing transferability as the training objective, it less clear how to get such pairs in practice, especially when training with real-world data.

To this end our third and final key insight is that given any sequence of frames $\mathcal{I}$, any two frame-wise augmentations AUG and AUG that *preserve the ground-truth camera pose*, *i.e.,*

$$\text{ORACLE}\left[\text{AUG}[\mathcal{I}]\right] = \text{ORACLE}\left[\text{AUG}[\mathcal{I}]\right] = \text{ORACLE}\left[\mathcal{I}\right], \tag{12}$$

can be used to produce two new sequences which share *identical* camera motion but very little pixel information. Combining this insight with the transferability objective gives rise to a novel training procedure. In practice, given an input pair we implement this strategy by randomly generating two equal-area masks whose union covers the whole image, and apply these in combination with color-jitter and blur to generate new pairs. Then, following the transferability objective in Equation (11), POSEENC extracts the relative pose latent from the first pair with which RENDER is asked to render the target image in the second pair given the first image as context.

We note that prior self-supervised methods RayZer and RUST also seek to prevent information leakage through different strategies. RayZer takes a more explicit approach by bottlenecking the pose latents via parameterization as elements of SE(3). However, as we show empirically in both benchmark comparisons and ablations, an explicit SE(3) parameterization not only fails to provide any degree of transferability in the multi-view setting, but in fact *degrades* it compared to an unconstrained stereo-monocular baseline. In contrast, RUST takes an approach that is more similar to our own wherein the target pose latent is estimated from the scene representation and a partial view of the target image, and the renderer asked to reconstruct the full target. However, RUST still suffers from interpolative bias via multi-view training and an augmentation strategy that does not eliminate much of the pixel content overlap.

### 3.5 XFACTOR: A MODEL FOR TRUE SELF-SUPERVISED NOVEL VIEW SYNTHESIS

Combining the [POSEENC, RENDER] *stereo-monocular* model with the *transferability objective* gives rise to XFactor (Figure 1) — short for Transferable[X] Latent Factorization[Factor]. As we demonstrate empirically in (Sec. 4.1), XFactor produces a fully transferable latent pose representation and to our knowledge is the first *fully self-supervised* model to achieve **true** NVS in the sense of Equations (5 – 6). We also note that XFactor does so *without any geometric or 3D inductive biases* whatsoever, demonstrating that such design choices are not a necessary condition for transferable latent poses.

Both POSEENC and RENDER are implemented as multi-view VITs. We define the image distance metric $d_I$ used to compute the transfer objective as a linear combination of the $L^1$ norm on the difference between the ground truth and predicted target image pixels and the LPIPS loss (Zhang et al., 2018), with a weight of $0.5$ on the latter. During training, augmentation masks are generated per batch example by splitting the patchified image plane into quadrants and randomly partitioning them into two groups of two. This not only allows for masks which either equally partition the image into left/right or upper/lower halves, but also diagonalized partitions. There also exists a small chance (5% in our implementation) that an image pair will not be masked, in which case transfer objective for that example reduces to the intra-sequence autoencoding objective and gives the model an opportunity to reason about the whole images. Augmentations are not applied during inference. Complete architectural and implementation details are included in Appendix A.

**Multi-View XFactor.** Given a trained XFactor stereo-monocular model we extend it to a multi-view model by fine-tuning [POSEENC, RENDER] in a secondary training stage. Here each multi-frame sequence $\mathcal{I} = \mathcal{I}_C \cup \{I_T\}$ is split into two disjoints sets consisting of context images and a single target image, with the latter chosen randomly. The reference image $I_R \in \mathcal{I}_C$ represents the view relative to which all poses will be expressed, and is chosen to be the frame with the minimum maximal baseline between all other frames (*i.e.* the "middlest"). We continue to use pose-preserving augmentations that are, however, now applied to all frames. For each sequence, POSEENC is applied pair-wise to predict the relative pose latents between the reference frame and all others. Then, RENDER is asked to render the target image of the second sequence with the second's context frames and poses and the target pose from the first.

## 4 EXPERIMENTS

We provide empirical evidence that XFactor produces a transferable camera pose representation (4.1) which well-predicts oracle SE(3) poses when probed (4.2). Last, we show through abla-

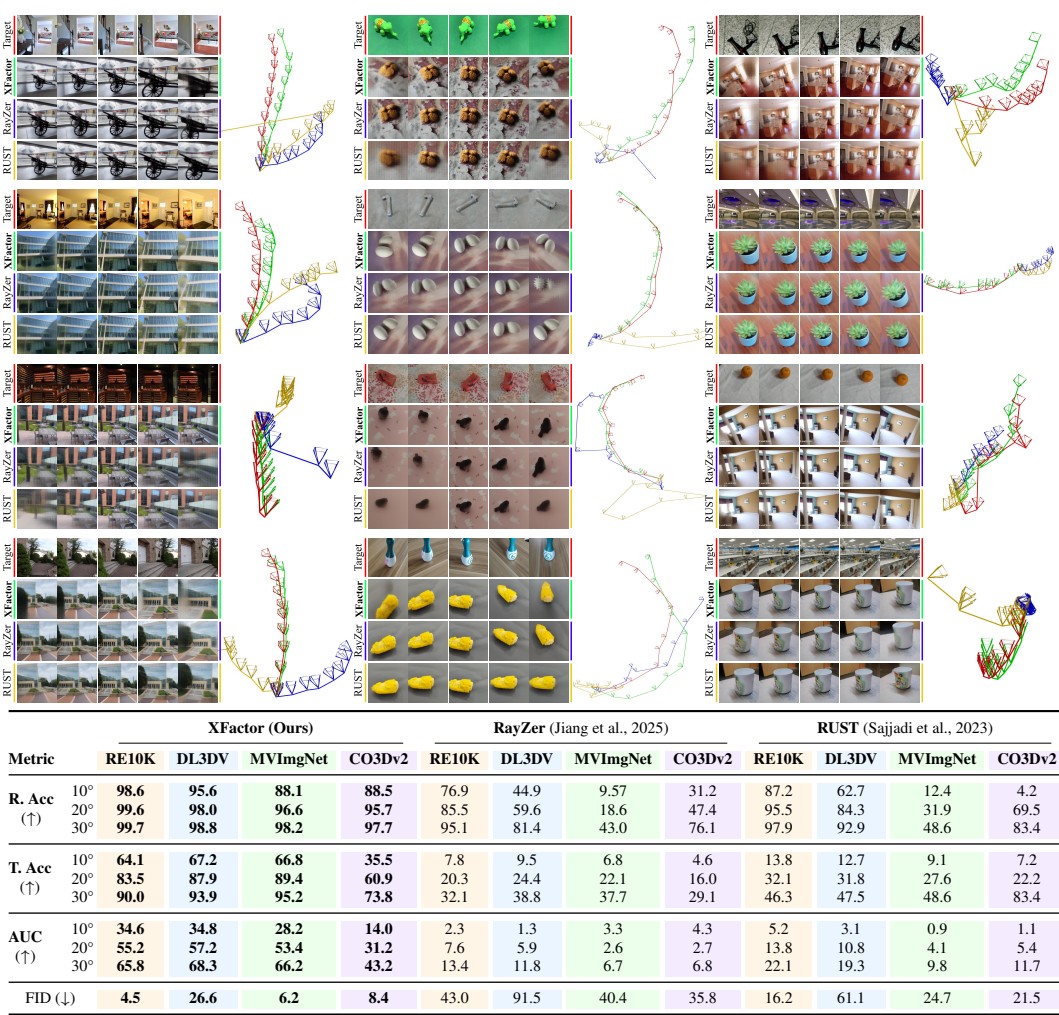

| Metric | **XFactor (Ours)** | | | | **RayZer** (Jiang et al., 2025) | | | | **RUST** (Sajjadi et al., 2023) | | | |
|---|---|---|---|---|---|---|---|---|---|---|---|---|
| | RE10K | DL3DV | MVImgNet | CO3Dv2 | RE10K | DL3DV | MVImgNet | CO3Dv2 | RE10K | DL3DV | MVImgNet | CO3Dv2 |
| **R. Acc** 10° | **98.6** | **95.6** | **88.1** | **88.5** | 76.9 | 44.9 | 9.57 | 31.2 | 87.2 | 62.7 | 12.4 | 4.2 |
| (↑) 20° | **99.6** | **98.0** | **96.6** | **95.7** | 85.5 | 59.6 | 18.6 | 47.4 | 95.5 | 84.3 | 31.9 | 69.5 |
| 30° | **99.7** | **98.8** | **98.2** | **97.7** | 95.1 | 81.4 | 43.0 | 76.1 | 97.9 | 92.9 | 48.6 | 83.4 |
| **T. Acc** 10° | 64.1 | 67.2 | 66.8 | 35.5 | 7.8 | 9.5 | 6.8 | 4.6 | 13.8 | 12.7 | 9.1 | 7.2 |
| (↑) 20° | 83.5 | 87.9 | 89.4 | 60.9 | 20.3 | 24.4 | 22.1 | 16.0 | 32.1 | 31.8 | 27.6 | 22.2 |
| 30° | 90.0 | 93.9 | 95.2 | 73.8 | 32.1 | 38.8 | 37.7 | 29.1 | 46.3 | 47.5 | 48.6 | 83.4 |
| **AUC** 10° | **34.6** | **34.8** | **28.2** | **14.0** | 2.3 | 1.3 | 3.3 | 4.3 | 5.2 | 3.1 | 0.9 | 1.1 |
| (↑) 20° | **55.2** | **57.2** | **53.4** | **31.2** | 7.6 | 5.9 | 2.6 | 2.7 | 13.8 | 10.8 | 4.1 | 5.4 |
| 30° | **65.8** | **68.3** | **66.2** | **43.2** | 13.4 | 11.8 | 6.7 | 6.8 | 22.1 | 19.3 | 9.8 | 11.7 |
| FID (↓) | **4.5** | **26.6** | **6.2** | **8.4** | 43.0 | 91.5 | 40.4 | 35.8 | 16.2 | 61.1 | 24.7 | 21.5 |

Table 1: **The Transferability Test.** We compare the transferability of XFactor's, RayZer's, and RUST's pose representations across four datasets. We evaluate using TPS with RRA, RTA, and AUC at different error thresholds. For all metrics except FID, higher is better. Visualizations of transfer renderings and camera trajectories extracted with ORACLE are shown for each method above. The target trajectory is visualized in red, XFactor in green, RayZer in blue, and RUST in gold.

tions that combining the stereo-monocular model with our transferability objective is ideal for both achieving transferability and producing a pose representation relative to a variety of alternative design decisions (4.3). In addition, we also report results on the benchmark of auto-encoding video sequences established by Jiang et al. (2025) in Appendix B.

**Comparisons.** We compare XFactor against two strong existing self-supervised NVS models: RayZer (Jiang et al., 2025) the current self-supervised state-of-the-art, and RUST (Sajjadi et al., 2023) which also estimates poses from partial views and does not make use of any 3D inductive bias. At this time, neither the authors of RayZer nor RUST have published code. We implement both models following the respective papers and shared our RayZer implementation with the authors, who confirmed it is faithful. For RUST, we view their principal contribution as predicting poses between *full* and *partial* views. Our implementation differs slightly from that of the authors: we do not leverage a set-latent scene representation, instead absorbing the scene encoder into a multi-view POSEENC and RENDER; POSEENC sees the set of all context views as well as the partial target view. Thus, comparisons against RUST can be seen as a comparison with training a multi-view model end-to-end with a full-to-partial objective instead of transferability.

**Datasets and Training.** We train all models on a large-scale, aggregate dataset consisting of real-world videos at both scene and object levels. Specifically, our dataset consists of RE10K (Zhou et al.,

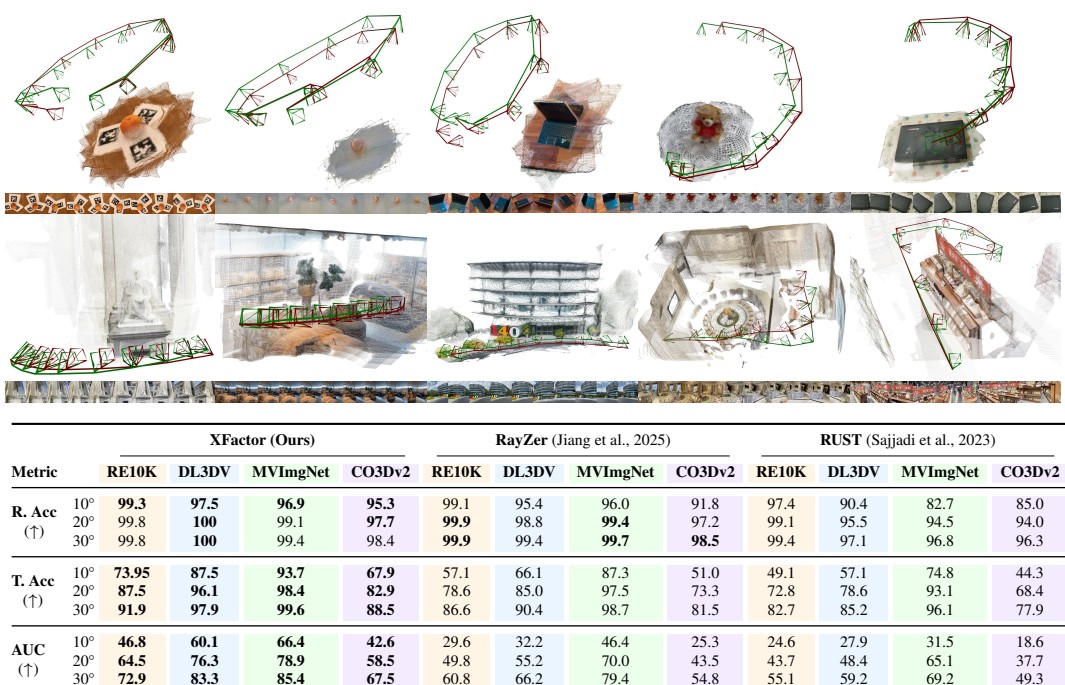

| Metric | | XFactor (Ours) | | | | RayZer (Jiang et al., 2025) | | | | RUST (Sajjadi et al., 2023) | | | |
|---|---|---|---|---|---|---|---|---|---|---|---|---|---|
| | | RE10K | DL3DV | MVImgNet | CO3Dv2 | RE10K | DL3DV | MVImgNet | CO3Dv2 | RE10K | DL3DV | MVImgNet | CO3Dv2 |
| R. Acc (↑) | 10° | **99.3** | **97.5** | **96.9** | **95.3** | 99.1 | 95.4 | 96.0 | 91.8 | 97.4 | 90.4 | 82.7 | 85.0 |
| | 20° | 99.8 | **100** | 99.1 | **97.7** | **99.9** | 98.8 | **99.4** | 97.2 | 99.1 | 95.5 | 94.5 | 94.0 |
| | 30° | 99.8 | **100** | 99.4 | 98.4 | **99.9** | 99.4 | **99.7** | **98.5** | 99.4 | 97.1 | 96.8 | 96.3 |
| T. Acc (↑) | 10° | 73.95 | 87.5 | 93.7 | 67.9 | 57.1 | 66.1 | 87.3 | 51.0 | 49.1 | 57.1 | 74.8 | 44.3 |
| | 20° | 87.5 | 96.1 | 98.4 | 82.9 | 78.6 | 85.0 | 97.5 | 73.3 | 72.8 | 78.6 | 93.1 | 68.4 |
| | 30° | 91.9 | 97.9 | 99.6 | 88.5 | 86.6 | 90.4 | 98.7 | 81.5 | 82.7 | 85.2 | 96.1 | 77.9 |
| AUC (↑) | 10° | 46.8 | 60.1 | 66.4 | 42.6 | 29.6 | 32.2 | 46.4 | 25.3 | 24.6 | 27.9 | 31.5 | 18.6 |
| | 20° | 64.5 | 76.3 | 78.9 | 58.5 | 49.8 | 55.2 | 70.0 | 43.5 | 43.7 | 48.4 | 65.1 | 37.7 |
| | 30° | 72.9 | 83.3 | 85.4 | 67.5 | 60.8 | 66.2 | 79.4 | 54.8 | 55.1 | 59.2 | 69.2 | 49.3 |

Table 2: **Pose Probe Accuracy.** We report probe accuracy trained to predict ground-truth SE(3) poses from the latents of each model in terms of RRA, RTA, and AUC. We show several examples of XFactor's poses (green) relative to ORACLE ground-truth (red). Zoom in to see details.

2018), DL3DV (Ling et al., 2024), MVImgNet (Yu et al., 2023), and CO3Dv2 (Reizenstein et al., 2021). Frames are first center-cropped and then resized to $256 \times 256$ pixels. Complete training and implementation details can be found in Appendix B.

## 4.1 TRANSFERABILITY

Our principal evaluation concerns transferability. Here, we compare multi-view XFactor, RayZer, and RUST on the evaluation splits of our aggregate dataset: For each dataset, we randomly draw 4000 pairs of sequences, select five equally spaced target frames, and compute the TPS as in Equation (9) with respect to RRA, RTA, and AUC at 10° intervals. We also compute the Frechet Inception Distance (FID) between the statistics of the sequences of input and rendered target frames as a general measure of transferred rendering quality.

The results, averaged over all sequences per dataset, and qualitative comparisons are shown in Table 1. XFactor significantly outperforms the other methods, reporting an AUC @ 20° over five times that of RayZer and RUST. Notably, despite sometimes producing reasonable looking renderings, both RayZer and RUST completely fail the transferability test and are not capable of *true* NVS. This is likely due to the susceptibility of their design toward learning interpolation latents due to end-to-end multi-view training with autoencoding. Of the two, RUST's performance is slightly better. We attribute this to its strategy of estimating pose latents between full and partial views as it brings its objective to a place between transferability and autoencoding.

## 4.2 POSE PROBE

Next, we evaluate the degree to which each model's latent pose representation encodes information about the ORACLE camera poses. To do so, we freeze the POSEENC from each model and train a three-layer MLP to predict the ground-truth SE(3) camera poses extracted by ORACLE from the estimated pose latents. We evaluate the quality of the probe-extracted trajectories relative to the ground-truth in terms of RRA, RTA, and AUC with the results shown in Table 2. Visualizations of poses extracted from XFactor's representations are shown above.

Overall, XFactor's pose latents provide a superior characterization of the oracle poses, with high AUC values at 10° and 20° and outperforming the other methods significantly. It follows that our

| | Metric | XFactor (Ours) | | | | Bottleneck | | | | Unconstrained | | | |
|---|---|---|---|---|---|---|---|---|---|---|---|---|---|
| | | RE10K | DL3DV | MVImgNet | CO3Dv2 | RE10K | DL3DV | MVImgNet | CO3Dv2 | RE10K | DL3DV | MVImgNet | CO3Dv2 |
| Transfer | R 20° (↑) | **99.9** | **98.36** | 96.5 | **98.2** | **99.9** | 97.7 | **96.7** | 98.1 | 99.8 | 98.0 | 95.3 | 97.9 |
| | T 20° (↑) | **75.1** | **76.6** | 84.1 | 33.05 | 70.4 | 72.6 | **85.3** | 35.5 | 64.3 | 70.4 | 81.0 | 31.99 |
| | AUC 20° (↑) | **47.2** | **44.8** | 49.3 | 14.6 | 40.6 | 41.4 | 50.8 | 15.8 | 36.8 | 39.4 | 47.0 | 14.0 |
| | FID (↓) | 3.40 | 34.7 | 7.74 | 6.14 | 3.29 | 33.9 | **6.79** | **5.56** | **3.26** | **31.7** | 6.95 | 5.80 |
| Probe | R 20° (↑) | **99.5** | **98.5** | **99.7** | **97.5** | 99.2 | 96.9 | 99.4 | 96.7 | 99.3 | 97.6 | 99.5 | 97.1 |
| | T 20° (↑) | **79.4** | **89.2** | **96.1** | **77.4** | 74.3 | 83.0 | 95.5 | 74.4 | 76.7 | 85.4 | 95.8 | 75.4 |
| | AUC 20° (↑) | **54.8** | **61.2** | **71.5** | **50.3** | 47.2 | 51.9 | 71.0 | 48.6 | 49.8 | 51.2 | 70.8 | 48.4 |

| | Metric | SE(3) & Plücker (Jiang et al., 2025) | | | | Additional View: Decoder | | | | Additional View: Encoder + Decoder | | | |
|---|---|---|---|---|---|---|---|---|---|---|---|---|---|
| | | RE10K | DL3DV | MVImgNet | CO3Dv2 | RE10K | DL3DV | MVImgNet | CO3Dv2 | RE10K | DL3DV | MVImgNet | CO3Dv2 |
| Transfer | R 20° (↑) | 99.8 | 94.2 | 0.826 | 95.6 | 99.7 | 94.2 | 86.1 | 97.7 | 99.6 | 93.9 | 43.5 | 96.2 |
| | T 20° (↑) | 65.3 | 58.6 | 70.7 | 31.7 | 62.0 | 52.2 | 53.1 | 35.3 | 18.2 | 19.7 | 14.2 | 10.4 |
| | AUC 20° (↑) | 35.7 | 26.4 | 28.4 | 12.6 | 33.1 | 25.1 | 22.3 | 14.9 | 7.2 | 6.3 | 1.2 | 3.2 |
| | FID (↓) | 3.53 | 36.1 | 7.37 | 6.11 | 5.77 | 50.5 | 16.2 | 9.86 | 5.0 | 36.6 | 6.4 | 9.1 |
| Probe | R 20° (↑) | 99.0 | 98.2 | 99.1 | 96.2 | 98.6 | 96.1 | 98.8 | 95.8 | 98.3 | 95.4 | 95.3 | 92.2 |
| | T 20° (↑) | 77.9 | 85.0 | 95.4 | 72.8 | 74.7 | 78.4 | 94.3 | 70.6 | 73.9 | 88.4 | 93.9 | 74.8 |
| | AUC 20° (↑) | 50.6 | 56.3 | 67.7 | 45.2 | 48.0 | 47.4 | 64.2 | 42.6 | 47.4 | 58.1 | 65.9 | 48.6 |

Table 3: **Ablations.** We ablate potential alternative design decisions using stereo-monocular XFactor as a starting point. Models are compared in terms of transferability and pose probe efficacy.

stereo-monocular model combined with our transferability objective also doubles as an effective method for self-supervised representation learning of 3D camera pose information. However, unlike the transferability test, neither RayZer nor RUST completely fail and in fact both learn a reasonably informative representation. This suggests that while transferability can improve geometric reasoning, evidence of the latter does not automatically lead to the former.

## 4.3 ABLATIONS

Here we ablate the influence of the fundamental components of XFactor's design — the stereo-monocular model and transferability objective. To do so, we train and evaluate several alternative models each representing different potential design decisions, and compare against stereo-monocular XFactor in terms of transferability and pose probe accuracy.

To ablate the influence of a stereo-monocular model relative to multi-view, we train two additional models with the transferability objective: A stereo POSEENC which uses a single additional context view in RENDER (Additional View: Decoder), essentially our proposed multi-view XFactor trained end-to-end: A full multi-view model wherein both POSEENC and RENDER see the additional context view (Additional View: Encoder + Decoder). To evaluate the effectiveness of the transferability objective we train three stereo-monocular models all with the standard autoencoding objective: One without any additional modification (Unconstrained); One with 16-dimensional pose latents, representing a bottleneck relative to the 256-dimensional latents used in XFactor (Bottleneck); One which predicts SE(3) poses and camera intrinsics and uses Plücker embeddings in the decoder following Jiang et al. (2025) (SE(3) & Plücker).

The results are shown in Table 3. While XFactor overall performs best out of all models, we see that transitioning to multi-view training, first by adding only an additional context view to RENDER, and then by adding a context view to POSEENC, progressively degrades, then completely destroys transferability. In contrast, the bottlenecked stereo-monocular model performs competitively with XFactor in terms of transferability, though XFactor's latents provide a comprehensively stronger characterization of real-world pose. However, we note that a bottlenecking strategy may not always be desirable and can limit descriptiveness, for instance, if one seeks a representation that also encodes changes in lighting or other evolving phenomena in a scene. In contrast, the transferability objective improves transferability without an explicit design constraint. Counterintuitively, we find that asking the stereo-monocular model to predict explicit SE(3) poses and camera parameters in fact significantly degrades transferability relative to both XFactor and the unconstrained baseline.

## 5 DISCUSSION

**Limitations.** While our model has claim to being the first geometry-free, fully-self supervised method to achieve true NVS, several limitations are outstanding. First, the restriction of POSEENC to a stereo model precludes ultra-wide baseline pose estimation in a single forward pass. In prin-

ciple, a multi-view POSEENC can robustly estimate latent poses across arbitrary baselines as long as it is possible to chain together a trajectory between frames that share overlap. In fact, such a model is highly effective in the supervised regime (Wang et al., 2025b), however, applying it in a self-supervised setting without introducing interpolative bias remains an open problem. Second, the rendering quality of transferred frames can exhibit blurring and warping artifacts which increase in frequency as the target poses diverge from those of the context. As illustrated in Fig. 3, deterministic NVS models supervised with ground-truth poses exhibit similar issues. Therefore, we posit this stems from the fact that XFactor is a deterministic, rather than generative, model and that the artifacts are a result of the model trying to resolve uncertainty without currently being equipped with the proper tools to do so. We believe that integrating recent advances in camera-controllable generative models (Song et al., 2025; Bai et al., 2025) into our approach could effectively address this limitation.

**Conclusion.** We have presented a new characterization of NVS that does not rely on notions from conventional multi-view geometry, instead formulating it as a pure machine-learning task in the form of a latent variable model. We identified transferability as the key input-output behavior of NVS. Based on our analysis, we introduced XFactor, the first geometry-free self-supervised model capable of true novel view synthesis. We provided large-scale experiments on real-world datasets that demonstrate XFactor's efficacy, and validate key design decisions with careful ablation studies. We hope that our analysis will encourage the community to seek new formulations of classic 3D vision problems based on key principles of machine learning.

## ACKNOWLEDGMENTS

Vincent Sitzmann was supported by the National Science Foundation under Grant No. 2211259, by the Singapore DSTA under DST00OECI20300823 (New Representations for Vision, 3D Self-Supervised Learning for Label-Efficient Vision), by the Intelligence Advanced Research Projects Activity (IARPA) via Department of Interior/Interior Business Center (DOI/IBC) under 140D0423C0075, by the Amazon Science Hub, by the MIT-Google Program for Computing Innovation, by Sony Interactive Entertainment, and by a 2025 MIT ORCD Seed Fund Compute Grant.

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

## A  XFACTOR: ARCHITECTURE AND IMPLEMENTATION DETAILS

The XFactor POSEENC and RENDER modules are implemented as multi-view ViTs with RoPE positional embeddings (Su et al., 2023). Following VGGT (Wang et al., 2025b), we fuse global and per-image attention inside each layer. While initially trained in the stereo-monocular setting, both POSEENC and RENDER are capable of handling an arbitrary number of views independent of the weights, though we only take advantage of this capability for RENDER when extending to multi-view. During training, both POSEENC and RENDER are passed a binary attention mask encoding the randomly generated disjoint partitions (AUG, AUG) which allows for encoding and rendering with respect to the transferability objective to be computed in a single forward pass.

POSEENC consists of local-global attention layers, followed by a pose head in the form of an MLP. A single global token is initialized and copied across the context and target view, representing the context and target pose latents. The attention layers are equivariant under swapping of the context and target image. Symmetry is broken by the pose head, which is designed such that the reference pose is always mapped to the zero vector.

RENDER is implemented similarly, consisting of local-global attention layers and an MLP pixel prediction head. The input pose latents are broadcasted across the token dimension, with the context pose latents fused to the pacification of the target image and the result is concatenated along the token dimension with the broadcasted target latents to form an internal two-view representation. After applying the attention layers, the features corresponding to the broadcasted target latents are extracted from the position of second image and passed to the pixel prediction head.

## B  EVALUATION

**Comparisons and Training.**  In our comparisons, we initialize all of XFactor's, RayZer's, and RUST's POSEENC, SCENEENC (used only in RayZer), and RENDER modules with eight transformer layers, 1024 features, 16 heads, and a patch size of 16, resulting in 16, 24, and 16 total layers for each model, respectively. Both XFactor and RUST use a latent pose dimension of 256. We standardize comparisons such that all methods render with 5 context views. Multi-view XFactor and RUST each take the reference view as one of the context views and along with a single additional target view. For RayZer, we use an additional 5 target views as is done in (Jiang et al., 2025).

We train with two separate baselines, with one set used in both training stereo-monocular XFactor and ablations, and the other used for both training multi-view XFactor, RayZer, RUST and all comparison evaluations. For the former, pairs of images are formed by randomly sampling frames from each dataset up to maximum baseline consisting of 100, 12, 12, and 20 frames for RE10K, DL3DV, MVImgNet, and CO3Dv2, respectively. These baselines were heuristically selected based on dataset difficulty and to ensure at least a small amount of overlap between pairs of frames. For fine-tuning, and training RayZer and RUST, we simply double the stereo-monocular baseline.

We train stereo-monocular XFactor, RayZer, and RUST all with the AdamW optimizer (Kingma & Ba, 2014), using a batch size of 256, weight decay of $5.0 \times 10^{-3}$, and a learning rate of $4.0 \times 10^{-4}$ for 100,000 iterations, decaying to $1.0 \times 10^{-4}$ on a cosine schedule. To extend XFactor to multi-view we fine-tune for an additional 100,000 iterations.

| Metric | XFactor (Ours) | | | | RayZer (Jiang et al., 2025) | | | | RUST (Sajjadi et al., 2023) | | | |
|---|---|---|---|---|---|---|---|---|---|---|---|---|
| | RE10K | DL3DV | MVImgNet | CO3Dv2 | RE10K | DL3DV | MVImgNet | CO3Dv2 | RE10K | DL3DV | MVImgNet | CO3Dv2 |
| PSNR (↑) | **26.1** | **23.2** | **24.8** | **26.5** | 22.9 | 20.3 | 21.4 | 22.6 | 18.9 | 17.7 | 19.1 | 19.3 |
| SSIM (↑) | **0.859** | **0.766** | **0.733** | **0.821** | 0.809 | 0.676 | 0.622 | 0.728 | 0.71 | 0.571 | 0.593 | 0.662 |
| LPIPS (↓) | **0.114** | **0.157** | **0.194** | **0.1677** | 0.135 | 0.188 | 0.258 | 0.205 | 0.220 | 0.286 | 0.3561 | 0.290 |

Table 4: **Autoencoding Reconstruction Quality.**

| Metric | XFactor (Aug. at Inference) | | | |
|---|---|---|---|---|
| | RE10K | DL3DV | MVImgNet | CO3Dv2 |
| PSNR (↑) | 26.1 | 22.7 | 24.2 | 25.5 |
| SSIM (↑) | 0.871 | 0.771 | 0.724 | 0.809 |
| LPIPS (↓) | 0.107 | 0.151 | 0.196 | 0.172 |
| FID (↓) | 2.55 | 22.1 | 3.43 | 4.84 |

Table 5: **Augmentations at Inference.** We evaluate transferred rendering quality in terms of standard perceptual metrics by applying our pose-preserving augmentations at inference with multi-view XFactor.

We train stereo-monocular XFactor on 4 NVIDIA H200 GPUs, taking approximately 15 hours to reach 100,000 iterations. Multi-view XFactor is trained on 8 NVIDIA H200 GPUs, taking approximately 48 hours to reach 100,000 iterations.

The pose probe is a three-layer MLP with a feature dimension of 256. The probe is trained for 10,000 iterations with a batch size of 64. The experiments differ slightly between the comparisons in Sec. 4.2 and ablations in Sec. 4.3. In the comparisons, the probe is used to predict five poses in a trajectory, and a scale-invariant loss is applied between the predicted and ORACLE-extracted trajectories. In the ablations, the probe is used to predict the relative pose between two frames and the loss is computed directly between the predicted pose and that of the ORACLE.

**Autoencoding Reconstruction.** Here we report results on autoencoding reconstruction for each model. This is simply the ability to render target frames from a sequence using scene and pose representations from the *same* sequence. While this work argues that this task is *fundamentally not equivalent* to true NVS and is purely a measure of a model's ability to act an interpolator, it is the principal evaluation benchmark used in RayZer and also provides a quantitative measure of reconstruction quality so we include it for completeness.

Autoencoding reconstruction results are shown in Table 4 in terms of standard perceptual metrics including PSNR, SSIM, and LPIPS averaged across sequences from each dataset. In this setting XFactor and RayZer achieve good reconstruction quality. We note that our results here differ slightly from those in the original paper. This is largely because we trained all models with a unified loss of $L^1$ with LPIPS, whereas the original RayZer uses $L^2$ with a custom VGG perceptual loss. This change improves LPIPS but lowers PSNR/SSIM, since we explicitly penalize LPIPS but not $L^2$ error. Additional discrepancies stem from training RayZer on our large-scale, mixed scene- and object-level dataset and using only 5 context views for fair comparisons across methods; the original RayZer trains separate models per dataset and uses more context views for several settings. These differences make our evaluation more demanding and naturally produce performance shifts.

**Perceptual Measure of Transfer Rendering.** To get a fuller picture of transfer rending quality, we evaluate multi-view XFactor by applying our pose-preserving augmentations at inference. Specifically, given a single sequence we apply our augmentations to generate two new sequences, extract the pose latents from each, and render the each sequence's target views with the others pose latents.

The results are shown in Table 5 where compare the transfer renderings against the ground truth target frames in terms of standard perceptual metrics. The results are very close to those achieved in autoencoding experiments in Table 4, demonstrating that XFactor can transfer with high fidelity when the distribution of target poses is close to those of the context used for rendering. However, this approach cannot be used to provide a fair comparison with RayZer and RUST, since both models do not see partially masked inputs during training and thus cannot be expected to well-handle them

| | XFactor (Ours) | | | | SimCLR (Chen et al., 2020) | | | | VICReg (Bardes et al., 2022) | | | |
|---|---|---|---|---|---|---|---|---|---|---|---|---|
| Metric | RE10K | DL3DV | MVImgNet | CO3Dv2 | RE10K | DL3DV | MVImgNet | CO3Dv2 | RE10K | DL3DV | MVImgNet | CO3Dv2 |
| **Probe** R 20° (↑) | **99.5** | **98.5** | **99.7** | **97.5** | 92.2 | 90.0 | 37.5 | 59.1 | 99.2 | 90.0 | 37.6 | 59.2 |
| T 20° (↑) | **79.4** | **89.2** | **96.1** | **77.4** | 12.5 | 7.6 | 1.5 | 4.5 | 2.6 | 0.0 | 0.1 | 1.5 |
| AUC 20° (↑) | **54.8** | **61.2** | **71.5** | **50.3** | 3.8 | 2.3 | 0.5 | 0.8 | 0.8 | 0.0 | 0.0 | 3.4 |

Table 6: **Comparison with Self-Supervised Objectives: Pose Probe Accuracy.** Probe accuracy for representations produced by training with our transferability objective (results copied from Table 3), a contrastive SimCLR (Chen et al., 2020) objective, and a mutual-information based VICReg (Bardes et al., 2022) objective.

| | XFactor (Ours) | | | | Temporal Shift +1 | | | | Temporal Shift +2 | | | |
|---|---|---|---|---|---|---|---|---|---|---|---|---|
| Metric | RE10K | DL3DV | MVImgNet | CO3Dv2 | RE10K | DL3DV | MVImgNet | CO3Dv2 | RE10K | DL3DV | MVImgNet | CO3Dv2 |
| **Transfer** R 20° (↑) | **99.9** | **98.36** | **96.5** | **98.2** | 99.9 | 96.4 | 92.1 | 97.65 | 99.8 | 96.1 | 91.0 | 97.6 |
| T 20° (↑) | **75.1** | **76.6** | **84.1** | **33.05** | 72.4 | 66.4 | 70.8 | 25.8 | 71.3 | 63.0 | 66.8 | 23.8 |
| AUC 20° (↑) | **47.2** | **44.8** | **49.3** | **14.6** | 41.6 | 36.2 | 36.1 | 11.1 | 41.1 | 32.7 | 29.2 | 9.38 |
| FID (↓) | **3.40** | **34.7** | **7.74** | **6.14** | 4.09 | 39.7 | 12.3 | 7.87 | 4.54 | 42.3 | 13.5 | 8.84 |
| **Probe** R 20° (↑) | **99.5** | **98.5** | **99.7** | **97.5** | 99.2 | 96.9 | 98.0 | 95.4 | 98.9 | 96.0 | 97.4 | 94.4 |
| T 20° (↑) | **79.4** | **89.2** | **96.1** | **77.4** | 74.1 | 78.4 | 92.2 | 67.4 | 67.6 | 74.6 | 91.3 | 65.7 |
| AUC 20° (↑) | **54.8** | **61.2** | **71.5** | **50.3** | 44.6 | 47.0 | 55.7 | 37.5 | 38.8 | 40.3 | 55.2 | 36.6 |

Table 7: **Robustness of Transferability Objective.** We progressively corrupt our stereo-monocular transferability objective by forming pairs of sequences by sometimes applying a temporal offset to pairs of frames, instead of via our proposed augmentation strategy. XFactor results are copied from Table 3.

at inference. Continuing to improve TPS and to comprehensively evaluate transferability remain important outstanding problems.

**Comparison Against Self-Supervised Learning Objectives.** We compare training POSEENC with our transferability objective to popular existing self-supervised objectives. To do so, we two train versions of our POSEENC module without the transferability objective: In the first, we train with a contrastive SimCLR (Chen et al., 2020) and in the second we train with the mutual-information based VICReg (Bardes et al., 2022) objective. In both cases, we feed two augmentations POSEENC to produce pairs of positively corresponding pose latents. For the SimCLR-style objective, for each positive pair in the batch, we use all other pose latents as negative examples to compute the InfoNCE with a temperature of $\tau = 0.1$. For the VICReg objective, the variance and covariance are computed over the batch. We do not train a RENDER module because we do not use the transferability objective. These models are otherwise trained in the same manner as is done with all other ablations.

To compare pose representations, we train a pose probe and report the probe accuracy in Table 6. We find that the SimCLR and VICReg objectives perform far worse than our transferability objective (and also all other design ablations in Table 3), almost completely failing to extract meaningful pose information. This suggests that rendering may be a critical facilitator of the emergence of geometric reasoning. Despite these results, we believe our method can still be improved by incorporating ideas from previous work on self-supervised learning, and will pursue this in future work.

**Robustness of Transferability Objective.** We perform additional experiments to evaluate the robustness of the transferability objective. We observe that for any pair of frames $\{I_i, I_j\}$ with temporal indices $\{i, j\}$, it is likely that the relative viewpoint change between $\{I_i, I_j\}$ will be similar to the change between frames $\{I_{i+1}, I_{j+1}\}$, though not exactly equal. Similarly, the viewpoint change between $\{I_{i+2}, I_{j+2}\}$ may also be similar, but less so. Thus, we train two new versions of our stereo-monocular model by progressively corrupting the transferability objective with pairs of images whose relative poses are similar, but imperfect. In the first experiment, for each example in the batch, there is an equal chance that pairs of sequences will be generated by our proposed augmentation scheme or by shifting the input frames by a temporal offset of one frame — e.g. $\mathcal{I}^A = \{I_i, I_j\}$ and $\mathcal{I}^B = \{I_{i+1}, I_{j+1}\}$. In the second, we increase the degree of corruption, and there exists an equal chance pairs will be generated by either our augmentation scheme, a temporal shift of 1, or a temporal shift of 2. These models are trained in the same manner as all other ablations.

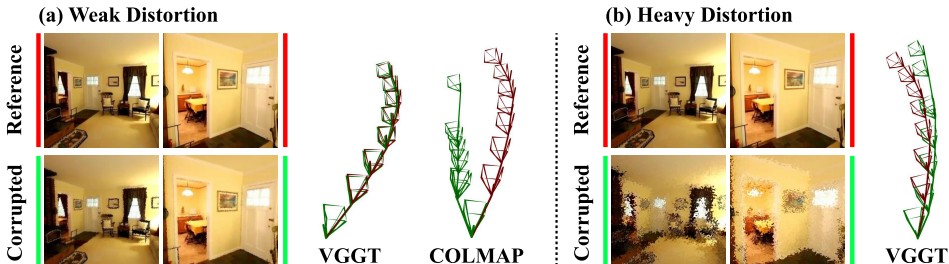

Figure 2: **Qualitative Robustness Analysis of TPS Oracles.** We evaluate the robustness of VGGT (Wang et al., 2025b) and COLMAP (Schönberger et al., 2016; Schönberger & Frahm, 2016b) oracles against visual corruptions. **(a)** VGGT is effectively invariant to minor distortions, while COLMAP quickly diverges. **(b)** VGGT maintains robustness even under significant random distortion, whereas COLMAP fails to register poses entirely.

| Metric | | VGGT Oracle | | | | COLMAP Oracle | | | |
|---|---|---|---|---|---|---|---|---|---|
| | | RE10K | DL3DV | MVImgNet | CO3Dv2 | RE10K | DL3DV | MVImgNet | CO3Dv2 |
| **R. Acc** (↑) | 10° | **100.0** | **100.0** | **100.0** | 99.2 | 86.3 | 96.4 | 88.9 | 91.2 |
| | 20° | **100.0** | **100.0** | **100.0** | 99.6 | 94.3 | 99.0 | 95.9 | 97.4 |
| | 30° | **100.0** | **100.0** | **100.0** | 99.7 | 97.3 | 99.6 | 97.8 | 99.3 |
| **T. Acc** (↑) | 10° | **87.7** | **97.0** | **96.9** | 82.5 | 38.7 | 71.5 | 58.6 | 44.3 |
| | 20° | **93.6** | **99.5** | **99.7** | 91.1 | 57.1 | 85.0 | 76.1 | 64.3 |
| | 30° | **95.7** | **99.9** | **99.9** | 93.7 | 69.7 | 90.0 | 84.1 | 74.1 |
| **AUC** (↑) | 10° | **70.4** | **80.5** | **74.5** | 61.1 | 20.2 | 46.9 | 35.7 | 24.3 |
| | 20° | **80.8** | **89.4** | **86.7** | 74.4 | 33.9 | 63.4 | 51.7 | 39.8 |
| | 30° | **85.4** | **92.8** | **91.1** | 80.4 | 43.6 | 71.5 | 61.2 | 49.7 |
| Success Rate (%) | | 100.0 | 100.0 | 100.0 | 100.0 | 28.9 | 52.3 | 21.4 | 32.0 |

Table 8: **Quantitative Robustness Analysis of TPS Oracles.** We evaluate the robustness of the VGGT (Wang et al., 2025b) and COLMAP (Schönberger et al., 2016; Schönberger & Frahm, 2016b) oracles against visual corruptions by measuring the TPS between reference and corrupted videos. A weak distortion, visualized in Fig. 2 (a), is applied to the video. We find that the VGGT oracle is significantly more robust across all datasets and metrics. In contrast, the COLMAP oracle suffers from an excessive rejection ratio, diminishing the utility of the evaluation samples.

The results are shown in Table 7. In terms of transferability, our objective proves remarkably robust. TPS falls off only slightly when corruption is first introduced and remains relatively stable as corruption increases while still outperforming all other ablative design decisions (Table 3) in terms of AUC @ 20°. However, probe accuracy degrades to a greater extent, suggesting imperfect viewpoint correspondence between sequences hampers the emergence of geometric reasoning.

**Robustness of TPS Oracles.** Any pose estimator can be used as a TPS oracle, but the evaluation quality may vary considerably based on this choice. To assess this variation, we compare the robustness of the VGGT oracle against the traditional SfM baseline, COLMAP (Schönberger et al., 2016; Schönberger & Frahm, 2016b) by using each method to compute the TPS between a given video and a corrupted version of the same video. We find that VGGT consistently yields a more robust TPS evaluation, as demonstrated qualitatively in Fig. 2 and quantitatively in Table 8. Additionally, we evaluate the COLMAP oracle on the actual transferred videos, with results provided in Table 9. In particular, the baseline RayZer (Jiang et al., 2025) suffers from excessively high rejection rate, making it impossible to reliably measure its transferability.

**Reconstruction Artifacts.** As shown in the visualizations in Tab. 1 and the videos on our website[2], XFactor's reconstruction is imperfect when rendering wide-baseline, out-of-sight, or out-of-

---

[2]https://www.mitchel.computer/xfactor/

Figure 3: **Supervised Methods Also Struggle.** Substantial reconstruction artifacts are also observed in supervised models when rendering difficult target views. The first and last frames are used as context views. **(a)** The supervised baseline, LVSM (Jin et al., 2024), excels at reconstructing easier target views. **(b)** However, LVSM struggles to cleanly render difficult target views, showing blurry or warped artifacts in the reconstruction.

| Metric | XFactor (Ours) | | | | RayZer (Jiang et al., 2025) | | | |
|---|---|---|---|---|---|---|---|---|
| | RE10K | DL3DV | MVImgNet | CO3Dv2 | RE10K | DL3DV | MVImgNet | CO3Dv2 |
| Success Rate (%) | 52.1 | 59.1 | 17.7 | 38.5 | 9.4 | 16.1 | 0.1 | 7.7 |

Table 9: **Fragility of COLMAP on Transferred Videos** We find that COLMAP is highly fragile when applied to transferred videos. In particular, samples from the baseline method, RayZer (Jiang et al., 2025), exhibit a significantly higher rejection ratio, preventing meaningful measurement of transferability with TPS.

distribution viewpoints. For instance, renderings of these difficult viewpoints often exhibit blurry or warped artifacts. However, we demonstrate that the supervised baseline, LVSM (Jin et al., 2024), suffers from similar issues, as illustrated in Fig. 3.

