# OpenReview forum: "True Self-Supervised Novel View Synthesis is Transferable"
_ICLR.cc/2026/Conference — ICLR 2026 Oral_

### Official Review · Reviewer_9vfk · 2025-10-29

**Soundness:** 3
**Presentation:** 4
**Contribution:** 2
**Rating:** 4
**Confidence:** 4

**Summary:**

This paper aims to define "true" self-supervised novel view synthesis (NVS) by introducing transferability as the key criterion. The paper defines transferability as the ability for a pose representation extracted from one video to render the same camera trajectory in a different scene. The paper further argues that prior generalizable, self-supervised NVS methods fail this test, as they learn to interpolate context frames rather than reason geometrically.

To solve this challenge, the paper presents XFactor, a model claimed to be "geometry-free" and capable of this "true" NVS. XFactor's design is based on two main ideas: 1) bootstrapping from a stereo-monocular (two-view) model to force extrapolation rather than interpolation, and 2) a new transferability objective that uses pose-preserving augmentations (e.g., inverse masking ) to disentangle pose from scene content. The authors also introduce a new metric, True Pose Similarity (TPS), to quantify this transferability. Experiments show XFactor significantly outperforms re-implementations of prior work (RayZer, RUST) on this new metric.

**Strengths:**

1. The paper identifies that previous generalizable self-supervised models (like RayZer and RUST) are prone to interpolating context frames when trained with a simple autoencoding objective, which is a valuable insight.

2. The transerability objective, enabled through the pose-preserving augmentation scheme, is a novel and original formulation for a self-supervised task. It provides a new way to force the model to learn a representation that disentangles the camera motion from the appearance.

3. The use of a stereo-monocular model to force extrapolation and avoid the interpolation failure mode is a clever, though costly, design choice.

4. The paper's internal logic is sound. The experiments rigorously support the paper's central, albeit narrow, claim: that the XFactor model, trained with its specific transferability objective, is superior at the task of pose-trajectory transfer, as measured by the proposed TPS metric. The ablations (Table 3) are particularly strong, effectively demonstrating that the novel objective is the key ingredient for this success, and that adding multi-view information from scratch is detrimental to this specific goal. The counterintuitive finding that an explicit $SE(3)$ bottleneck is harmful is genuinely interesting.

5. The paper is exceptionally well-written, clear, and logically structured. The authors do an excellent job of motivating their specific problem formulation, defining their terms (transferability, TPS), and explaining their method (XFactor). Figure 1 provides a very clear illustration of the novel training objective.

**Weaknesses:**

1. The paper's central premise—that "transferability" is the key criterion for "true NVS" —is an overstatement. The paper defines "true NVS" in a way that conveniently excludes the most dominant NVS paradigms, such as single-scene optimization (NeRF, 3D Gaussian Splatting), which the paper dismisses as "Oracle Methods". The work's scope is limited to generalizable, unposed video models, and the abstract and introduction must be revised accordingly.

2. The paper equates "transferability" with "user-controllable" NVS. The proposed XFactor model does not achieve this, though. A user cannot specify a novel viewpoint; they can only copy a viewpoint from another existing video sequence. The learned 256-dimensional latent pose is a black box. This is a "pose-copying" system, not a "pose-controlling" one, which is a significant gap between the stated problem and the delivered solution.

3. The claim of being "geometry-free" is a bit misleading in my opinion. The entire transferability objective is built on a strong, expert-provided geometric prior: that augmentations like masking are pose-preserving, a fact the authors even formalize as $ORACLE[AUG[\mathcal{I}]] = ORACLE[\mathcal{I}]$. This is not a "pure machine learning problem"; it is a clever re-encoding of a geometric bias into the training data.

4. The paper compares XFactor to re-implemented baselines (RayZer, RUST)  on its own novel metric (TPS), a metric for which XFactor is explicitly optimized. These baselines, unsurprisingly, fail spectacularly. The paper's own ablation (Table 3, "Unconstrained" baseline) shows that a standard stereo-monocular model with an autoencoding objective doesn't do much worse. This shows the win is not from a superior architecture but mainly from a training objective that is perfectly aligned with the evaluation metric, making the comparison to prior art feel circular and a bit unfair.

5. The model's reliance on a stereo-monocular POSEENC to avoid interpolation is a major tradeoff. It sacrifices the core strength of multi-view 3D reconstruction: robust integration of information across many frames. This design choice may solve the interpolation proble,m but it creates a model that is likely less robust and has a poorer understanding of 3D geometry than a true multi-view system.

**Questions:**

Given that your criterion of "transferability" does not apply to single-scene NVS (e.g., NeRF), will you concede that your paper is not about "true NVS" in general, but rather about the more specific sub-problem of generalizable NVS from unposed video?

You state your goal is a "user-controllable viewpoint". How can a user control the 256-dim latent pose vector of XFactor without first having a source video with the exact camera motion they desire? Isn't this a "pose-copying" system, not a "pose-controlling" one?

Your training objective simulates two sequences with identical poses. How brittle is the learned representation to this perfect "pose-matching" assumption? Have you analyzed the performance degradation when the source and context trajectories are not identical, but merely similar?

---

> ### Author Response · Authors · 2025-11-21
> **Official Comment by Authors [1/3]**
>
> We thank the reviewer for providing valuable feedback that helped us improve the clarity of our claims.
>
>
> ### [Concern 1] Single-scene Optimization methods are dismissed.
> > The paper's central premise—that "transferability" is the key criterion for "true NVS" —is an overstatement. The paper defines "true NVS" in a way that conveniently excludes the most dominant NVS paradigms, such as single-scene optimization (NeRF, 3D Gaussian Splatting), which the paper dismisses as "Oracle Methods". The work's scope is limited to generalizable, unposed video models, and the abstract and introduction must be revised accordingly.
>
> Yes, we agree completely. It was never our intention to say these models are not “true NVS models” and in fact we endeavor to claim the opposite: These supervised single-scene methods (and supervised feed-forward methods) are already true NVS models under the definition of transferability. Our use of the term “Oracle” is used to denote that these models are trained with ground-truth supervision.  We apologize for the misunderstanding, and have expanded the discussion in section 3.1 and 3.2 to make crystal clear that single-scene optimization methods such as NeRF and Gaussian Splatting also perform true novel view synthesis.
>
> More generally, our intent has been to draw attention to the fact that for existing state-of-the-art approaches (both feed-forward an single-scene) trained with oracle supervision, transferability arises almost trivially (Equation (7) in the paper). This is a fantastic benefit and we argue this is exactly what makes these models capable of “true NVS”. The point of our paper is that in the self-supervised case, there are no such guarantees of transferability, and special care must be taken to ensure it arises.
>
>
> ### [Concern 2] User controllabilty.
> > The paper equates "transferability" with "user-controllable" NVS. The proposed XFactor model does not achieve this, though. A user cannot specify a novel viewpoint; they can only copy a viewpoint from another existing video sequence. The learned 256-dimensional latent pose is a black box. This is a "pose-copying" system, not a "pose-controlling" one, which is a significant gap between the stated problem and the delivered solution.
>
>
> We appreciate the reviewer's feedback regarding the definition of controllability. We agree that XFactor is a "pose-copying" system; however, we posit that the pose-copying capability is a sufficient condition for user controllability. By synthesizing a reference video with a user-specified camera motion (e.g., via a game engine) and transferring it to the target scene, a user can control the camera trajectory. However, we acknowledge that this crucial discussion on the non-obvious connection between pose-copying and user controllability was missing in the original version of our paper. To clarify this "control-by-proxy" workflow, we have made a proof-of-concept video in which a synthetic target video rendered from a 3D simulator is transferred to real world scenes. Please find the video in our webpage: https://xfactor-transferable-nvs.github.io
>
> ### [Concern 3] Correctness of the term "geometry-free."
> > The claim of being "geometry-free" is a bit misleading in my opinion. The entire transferability objective is built on a strong, expert-provided geometric prior: that augmentations like masking are pose-preserving, a fact the authors even formalize as $Oracle[Aug[I]]=Oracle[I]$. This is not a "pure machine learning problem"; it is a clever re-encoding of a geometric bias into the training data.
>
> We thank the reviewer for the comment. We agree that “geometry-free” is an imprecise term with no clear meaning - however, we note that our use is consistent with the common use of this term in the novel view synthesis community, e.g. in the Scene Representation Transformer [1] and LVSM [2] which both claim “geometry-free view synthesis”. We have added a clarification to the paper that "We use the term `geometry-free' to denote that the model is free of heuristic 3D representations such as Gaussian splats, volumetric rendering or Plücker embeddings."
>
> [1] Sajjadi, Mehdi SM, et al. "Scene representation transformer: Geometry-free novel view synthesis through set-latent scene representations." Proceedings of the IEEE/CVF Conference on Computer Vision and Pattern Recognition. 2022.
>
> [2] Jin, Haian, et al. "Lvsm: A large view synthesis model with minimal 3d inductive bias." arXiv preprint arXiv:2410.17242 (2024).

---

> ### Author Response · Authors · 2025-11-21
> **Official Comment by Authors [2/3]**
>
> ### [Concern 4] Unfair Evaluation.
> > The paper compares XFactor to re-implemented baselines (RayZer, RUST) on its own novel metric (TPS), a metric for which XFactor is explicitly optimized. These baselines, unsurprisingly, fail spectacularly. The paper's own ablation (Table 3, "Unconstrained" baseline) shows that a standard stereo-monocular model with an autoencoding objective doesn't do much worse. This shows the win is not from a superior architecture but mainly from a training objective that is perfectly aligned with the evaluation metric, making the comparison to prior art feel circular and a bit unfair.
>
>
> We respectfully disagree that the comparison is unfair or circular. Our work addresses a fundamental question: "What constitutes a valid representation for NVS?" We built our approach on the first principle that a valid pose representation should always render the same view transformation regardless of the scene. Consequently, TPS is not a "made-up" metric designed to favor our method; it was desiged to bring the abstract notion of transferability into a quantifiable problem.
>
> The baselines (RayZer and RUST) claim self-supervised NVS, but our analysis reveals that they fail to learn a transferable representation. Instead, they rely on scene-specific shortcuts (entanglement) to minimize the autoencoding loss. Therefore, their failure on TPS is not a result of "unfair" optimization, but proof that they do not solve the fundamental NVS problem.
>
> Finally, regarding the reviewer's statement that XFactor is 'explicitly optimized' for TPS, we clarify that XFactor is never trained on TPS. The metric serves solely as a post-training evaluation tool.
>
>
>
>
>
> ### [Concern 5] Limitation of stereo pose encoder.
> > The model's reliance on a stereo-monocular POSEENC to avoid interpolation is a major tradeoff. It sacrifices the core strength of multi-view 3D reconstruction: robust integration of information across many frames. This design choice may solve the interpolation proble,m but it creates a model that is likely less robust and has a poorer understanding of 3D geometry than a true multi-view system.
>
>
> We agree, and explicitly acknowledge this as a limitation. We are not claiming that our solution is optimal, rather that is one way to solve this problem. It is not immediately clear how to achieve transferability with a self-supervised multi-view pose estimation module, and we agree that this is an important question for future work.

---

> ### Author Response · Authors · 2025-11-21
> **Official Comment by Authors [3/3]**
>
> ### [Questions]
>
> > Given that your criterion of "transferability" does not apply to single-scene NVS (e.g., NeRF), will you concede that your paper is not about "true NVS" in general, but rather about the more specific sub-problem of generalizable NVS from unposed video?
>
> Please review our response to **[Concern 1]**.
>
> > How can a user control the 256-dim latent pose vector of XFactor without first having a source video with the exact camera motion they desire? Isn't this a "pose-copying" system, not a "pose-controlling" one?
>
> A user can synthesize a video with the desired camera motion using a simulator. We have made a new proof-of-concept video to demonstrate this concept in our webpage:  https://xfactor-transferable-nvs.github.io
>
>
> > Your training objective simulates two sequences with identical poses. How brittle is the learned representation to this perfect "pose-matching" assumption? Have you analyzed the performance degradation when the source and context trajectories are not identical, but merely similar?
>
>
> We thank the reviewer for this constructive suggestion. To this point, we perform additional experiments to evaluate the robustness of the transferability objective.  We observe that for any pair of frames $\{I_i, I_j\}$ with temporal indices $\{i, j\}$, it is likely that the relative viewpoint change between $\{I_i, I_j\}$ will be similar to the change between frames $\{I_{i + 1}, I_{j + 1}\}$, though not exactly equal. Similarly, the viewpoint change between $\{I_{i + 2}, I_{j + 2}\}$ may also be similar, but less so. Thus, we train two new versions of our stereo-monocular model by progressively corrupting the transferability objective with pairs of images whose relative poses are similar, but imperfect. In the first experiment, for each example in the batch, there is an equal chance that pairs of sequences will be generated by our proposed augmentation scheme or by shifting the input frames by a temporal offset of one frame — e.g. $\mathcal{I}^A = \{I_i, I_j\}$ and $\mathcal{I}^B = \{I_{i + 1},  I_{j + 1}\}$. In the second, we increase the degree of corruption, and there exists an equal chance pairs will be generated by either our augmentation scheme, a temporal shift of 1, or a temporal shift of 2.
>
> The results are shown in Table 7 in the appendix. In terms of transferability, our objective proves remarkably robust. TPS falls off only slightly when corruption is first introduced and remains relatively stable as corruption increases while still outperforming all other ablative design decisions (Table 3) in terms of AUC @ 20°. However, probe accuracy degrades to a greater extent, suggesting imperfect viewpoint correspondence between sequences hampers the emergence of geometric reasoning.

---

> > ### Comment · Reviewer_9vfk · 2025-11-26
> > **Thank you for this great rebuttal**
> >
> > I appreciate the authors' exceptionally detailed and rigorous rebuttal. The responses have successfully addressed all my initial concerns, and the paper is significantly strengthened by the included clarifications and new experiments. I am satisfied with the clarification that the work focuses on achieving transferability in the self-supervised, generalizable setting, which clarifies the paper's scope against supervised methods. The sim-to-real transfer workflow with additional videos on the website provides a practical and sufficient path to user controllability, effectively addressing the gap between "pose-copying" and "pose-controlling." The new ablation experiments are highly convincing and significantly strengthen the paper's empirical foundation. I accept the argument that the failure of baselines on the TPS metric proves their reliance on interpolation, validating the importance of the proposed transferability criterion. The central contribution—redefining self-supervised NVS through transferability and providing a principled method (XFactor) to achieve it is also novel and impactful. The paper is exceptionally well-written, and the authors have been highly responsive. I have updated my confidence and rating from 4 to 8 and vote to accept this paper.

---

> > > ### Author Response · Authors · 2025-11-27
> > >
> > > We sincerely appreciate your detailed constructive feedback and your willingness to engage with us.  Your insightful criticisms helped us clarify key points and significantly improve the paper --- thank you for your time and effort!

---

### Official Review · Reviewer_VrW1 · 2025-11-01

**Soundness:** 3
**Presentation:** 4
**Contribution:** 3
**Rating:** 6
**Confidence:** 4

**Summary:**

This paper addresses the problem of self-supervised novel view synthesis (NVS). The authors make a crucial and insightful observation: the key criterion for a "true" NVS model is transferability, i.e., a representation of a camera pose extracted from one video sequence should be usable to render the exact same camera trajectory in a completely different scene. They argue that prior self-supervised, geometry-free methods fail this test, as their learned "pose" representations are entangled with scene content, leading them to act as sophisticated frame interpolators rather than genuine view synthesizers.

To address this, the paper introduces XFactor, a novel self-supervised NVS model designed explicitly to learn transferable pose representations. The authors quantify transferability by proposing TPS (True Pose Similarity), and prevent the model from learning to interpolate by making XFactor bootstrapped from a stereo-monocular model and trained with a pose-preserving data augmentation strategy. Experimental results demonstrate the effectiveness.

**Strengths:**

The paper asks a fundamental question in the field of self-supervised NVS. The proposed "transferability" is a powerful and original reframing of the problem. The paper is well-written and well-motivated. The design of the proposed solutions is elegant and principled, with its effectiveness validated by extensive experiments. Through the lens of transferability, the paper provides valuable insights to the community of self-supervised NVS.

**Weaknesses:**

I appreciate the paper's main idea and find its core contribution to be insightful and significant. While the work is very strong, I have some concerns, primarily regarding the evaluation methodology for what the paper defines as a "true NVS model".


### 1. The Challenge of Evaluating a True NVS Model

My main concern is how to fairly and comprehensively evaluate NVS performance under the new, important definition of "true NVS model". While I agree that standard autoencoding metrics like those in Table 4 are biased, I am not convinced that the proposed True Pose Similarity (TPS) metric (Eq. 9), used in Tables 1-3, provides a complete picture. My reasons are as follows:

**1.1. Assessing Perceptual Quality in the Transfer Setting:** The TPS metric evaluates the *geometric consistency* of a transferred camera trajectory but does not directly measure the *perceptual quality* of the synthesized images. The paper itself compellingly argues that autoencoding reconstruction (seq-to-seq rendering on the same scene) is "not equivalent to true NVS" (lines 641-642) and is merely a measure of interpolation ability. This creates a critical gap in the evaluation: we can see from the TPS scores in Table 1 & 2 that XFactor is geometrically consistent, but we have no quantitative perceptual measure of the actual view synthesis quality, which should be a key point for NVS methods. This leaves the assessment of the model's core capability incomplete.

**1.2. Fragility of the Oracle-based Metric:** The TPS metric's reliance on a post-hoc oracle (VGGT) makes it potentially fragile. As the paper notes (lines 199-202), the metric could be "hacked". More practically, it can fail in challenging extrapolation scenarios. For example, consider transferring a forward-motion trajectory from a driving scene (A) to a scene featuring an ego-centric rotation of an object (B). A model might produce meaningless, blurry artifacts for $\text{Render}(\mathcal{S}^B, \mathcal{Z}_T^A)$, as it is a difficult extrapolation synthesis task (line 474-477). In such a case, an oracle like VGGT would likely fail to extract any meaningful pose from these artifacts, leading to a poor TPS score. This low score, however, would be an artifact of the evaluation tool's failure, not necessarily a failure of the model's transferability logic.

**1.3. Inherent Noise in Oracle Tools:** It is well-documented that structure-from-motion oracles like COLMAP, and the feed-forward methods like VGGT, are not perfectly reliable and can produce noisy or incorrect poses. This issue has been analyzed in prior work such as RayZer [1] (arxiv version, Figure 4) and Less3Depend [2] (arxiv version, Appendix F). This unreliability introduces noise into the evaluation, making it difficult to disentangle the model's errors from the oracle's errors.

Given these points, I wonder if a well-defined perceptual measurement, independent of TPS, is necessary to complete the evaluation. For example, leveraging the **Pose-Preserving Augmentation** framework (Eq. 11) at **test time**. By applying two different augmentations to the same source sequence, the model could render a target frame from the second augmented sequence using the pose from the first. The resulting image could then be directly compared to the ground-truth target using standard perceptual metrics (PSNR, LPIPS, etc.). This would provide a direct, quantitative measure of rendering quality under the context of true transferability, which might complement the TPS metric.

I believe a discussion of these evaluation limitations in the main paper or appendix would be highly beneficial. On this point, I am open to hearing the authors' clarification and am highly willing to raise my rating if this concern is addressed.

### 2. Potentially Related Work

The paper astutely identifies the ill-posed nature of the autoencoding objective for NVS. This issue has also been noted by [3], which also addresses this problem, but proposes an alternative solution that involves leveraging explicit 3D representations for fine-tuning. Including a discussion that contrasts XFactor's geometry-free, transferability-based approach with this alternative 3D-aware solution would further strengthen the paper's contribution and contextualize it within the latest research landscape.


---


[1] RayZer: A Self-supervised Large View Synthesis Model (ICCV'25)

[2] The Less You Depend, The More You Learn: Synthesizing Novel Views from Sparse, Unposed Images without Any 3D Knowledge (arxiv)

[3] Recollection from Pensieve: Novel View Synthesis via Learning from Uncalibrated Videos (arxiv)

**Questions:**

1. The reported performance of the RayZer baseline in Table 4 appears to be a significant degradation when compared to the results presented in the original RayZer paper (Figure 1). I recognize and appreciate that the authors had to re-implement this baseline due to the lack of public code, and commend them for seeking confirmation from the original authors that their implementation is "faithful" (lines 312-314). However, the substantial performance gap is confusing. Are there key differences in the training dataset, evaluation protocol, or hyperparameter settings compared to those used in the original RayZer publication?

---

> ### Author Response · Authors · 2025-11-21
> **Official Comment by Authors [1/3]**
>
> We appreciate the reviewer for acknowledging the significance of our formulation. We are also grateful for the valuable feedback on our evaluation method. We have addressed all the concerns and questions below.
>
> ### [Concern 1] Perceptual Quality in the Transfer Setting Missing.
> > The TPS metric evaluates the geometric consistency of a transferred camera trajectory but does not directly measure the perceptual quality of the synthesized images. The paper itself compellingly argues that autoencoding reconstruction (seq-to-seq rendering on the same scene) is "not equivalent to true NVS" (lines 641-642) and is merely a measure of interpolation ability. This creates a critical gap in the evaluation: we can see from the TPS scores in Table 1 & 2 that XFactor is geometrically consistent, but we have no quantitative perceptual measure of the actual view synthesis quality, which should be a key point for NVS methods. This leaves the assessment of the model's core capability incomplete.
>
>
> >Given these points, I wonder if a well-defined perceptual measurement, independent of TPS, is necessary to complete the evaluation. For example, leveraging the Pose-Preserving Augmentation framework (Eq. 11) at test time. By applying two different augmentations to the same source sequence, the model could render a target frame from the second augmented sequence using the pose from the first. The resulting image could then be directly compared to the ground-truth target using standard perceptual metrics (PSNR, LPIPS, etc.). This would provide a direct, quantitative measure of rendering quality under the context of true transferability, which might complement the TPS metric.
>
>
> We agree that a perceptual measurement of transferred rendering quality is necessary to verify effective transferability. To this end, in Table 1 we included the FID score computed between the renderings of the transferred sequences and the context images in a given scene to give a measurement of the fidelity and faithfulness of the transferred renderings. However, we recognize that this analysis is incomplete and thank the reviewer for their helpful suggestion for an additional perceptual measure.
>
> To this end, we perform an additional evaluation with the results shown in Table 5 of the appendix. Here we apply our masked pose-preserving augmentations at inference exactly as suggested, using the pose latents from one masked region to render the other and vice versa. Then, we compute standard perceptual metrics between the transferred rendering and the input sequence.  The results are very close to those achieved in autoencoding experiments, demonstrating that XFactor can transfer with high fidelity when the distribution of target poses is close to those of the context used for rendering.
>
> However, this approach cannot be used to provide a fair comparison with RayZer and RUST, since both models do not see partially masked inputs during training and thus cannot be expected to well-handle them at inference. Continuing to improve TPS and to comprehensively evaluate transferability are important problems we plan to address in future work.

---

> ### Author Response · Authors · 2025-11-21
> **Official Comment by Authors [2/3]**
>
> ### [Concern 2] Fragility of the Oracle-based Metric.
> >  The TPS metric's reliance on a post-hoc oracle (VGGT) makes it potentially fragile. As the paper notes (lines 199-202), the metric could be "hacked". More practically, it can fail in challenging extrapolation scenarios. For example, consider transferring a forward-motion trajectory from a driving scene (A) to a scene featuring an ego-centric rotation of an object (B). A model might produce meaningless, blurry artifacts for $Render(S^B, Z_T^A)$, as it is a difficult extrapolation synthesis task (line 474-477). In such a case, an oracle like VGGT would likely fail to extract any meaningful pose from these artifacts, leading to a poor TPS score. This low score, however, would be an artifact of the evaluation tool's failure, not necessarily a failure of the model's transferability logic.
>
> > It is well-documented that structure-from-motion oracles like COLMAP, and the feed-forward methods like VGGT, are not perfectly reliable and can produce noisy or incorrect poses. This issue has been analyzed in prior work such as RayZer [1] (arxiv version, Figure 4) and Less3Depend [2] (arxiv version, Appendix F). This unreliability introduces noise into the evaluation, making it difficult to disentangle the model's errors from the oracle's errors.
>
>
> We clarify that TPS is not defined by the absolute correctness of the oracle's pose prediction. Instead, we designed TPS to measure the pose similarity perceived by the oracle. Therefore, TPS should not be interpreted as a transferability metric for individual scenes; rather, it is intended to compare aggregate statistics over the entire evaluation set. Consequently, noisy oracle predictions are not necessarily problematic and can even serve as important indicators of geometric inconsistency. We note that the issues with incorrect poses in [1,2] are discussed in the context of supervised training, which differs fundamentally from our evaluation problem.
>
>
> Nevertheless, we agree that the potential fragility of the TPS oracle is a valid concern. To address this, we conducted a robustness analysis of the VGGT oracle, detailed in Appendix B. As shown in Figure 2 and Table 8, the VGGT oracle exhibits strong robustness to visual corruptions.
>
> [1] RayZer: A Self-supervised Large View Synthesis Model (ICCV'25)
>
> [2] The Less You Depend, The More You Learn: Synthesizing Novel Views from Sparse, Unposed Images without Any 3D Knowledge (arxiv)
>
>
> &nbsp;
> ### [Concern 3] Related Works.
> > The paper astutely identifies the ill-posed nature of the autoencoding objective for NVS. This issue has also been noted by [3], which also addresses this problem, but proposes an alternative solution that involves leveraging explicit 3D representations for fine-tuning. Including a discussion that contrasts XFactor's geometry-free, transferability-based approach with this alternative 3D-aware solution would further strengthen the paper's contribution and contextualize it within the latest research landscape.
> [3] Recollection from Pensieve: Novel View Synthesis via Learning from Uncalibrated Videos (arxiv)
>
> We thank the reviewer for this pointer. We agree that this paper is highly relevant and have added a discussion to the related work section. As mentioned by the reviewer, our method differs from [3] in that we do not rely on explicit 3D modeling.

---

> ### Author Response · Authors · 2025-11-21
> **Official Comment by Authors [3/3]**
>
> ### [Questions]
> > The reported performance of the RayZer baseline in Table 4 appears to be a significant degradation when compared to the results presented in the original RayZer paper (Figure 1). I recognize and appreciate that the authors had to re-implement this baseline due to the lack of public code, and commend them for seeking confirmation from the original authors that their implementation is "faithful" (lines 312-314). However, the substantial performance gap is confusing. Are there key differences in the training dataset, evaluation protocol, or hyperparameter settings compared to those used in the original RayZer publication?
>
> Thank you for recognizing this point of clarification. We note that our reported results for RayZer show degradation in PSNR and SSIM but also an improvement in LPIPS compared to the results reported in the original RayZer paper. There are several factors which likely influence the difference in our reported results versus those reported in the original RayZer paper.
>
> To ensure fair comparisons, we trained all of our models with the same loss $d_I$, which differs from the one used in the original implementation of RayZer. Specifically, our loss is the sum of the $L^1$ loss (pixel-wise) with the LPIPS loss while RayZer uses the sum of the $L^2$ loss (pixel-wise) with a custom VGG perceptual loss. In practice, we found that our combination of $L^1$ + LPIPS produced (in our opinion) visually superior results when compared to the loss used in RayZer. This is likely the primary contributor to the difference in reported results. Our version of RayZer achieves a better LPIPS score on RE10K and DL3DV than what is reported in the original paper (lower is better, 0.135 and 0.188 vs 0.164 and 0.222, respectively) as we explicitly penalize for it in our loss term whereas RayZer does not. In tandem, our version also scores lower on PSNR on both datasets (higher is better, 22.9 and 20.3 vs 26.3 and 23.72) likely because PSNR is a function of the $L^2$ metric, which we do not explicitly penalize but the original implementation of RayZer does.
>
> There are other differences between our implementations which also likely contribute to the discrepancy. In this paper, we train RayZer on our large-scale combined dataset which includes all of RE10K and DL3DV (scene-level videos) and MVImgNet and CO3Dv2 (object-level videos). RayZer trains a different model per dataset, and does not report results on any real-world object level videos. In our regime, RayZer is forced to build representations that are effective across both scene and object levels instead of specializing in a single paradigm, so it would be reasonable to expect some fall-off on individual datasets as the model tries to generalize. Furthermore, for the sake of fair comparisons with XFactor and RUST, we train all models with the same number of context views (5) and so train RayZer with 5 context views and 5 target views which is one regime proposed by the authors in the paper. While this regime is used in the original paper for training and evaluation on RE10K, 16 context/8 target and 12 context/8 target are used to train RayZer on DL3DV and at object-level, respectively, in the original paper. Having fewer context views increases uncertainty, so our experiments can be viewed as more challenging than those in the original paper.

---

> > ### Comment · Reviewer_VrW1 · 2025-11-25
> >
> > Thank you for the detailed rebuttal. Regarding (W1.1) Perceptual Quality Evaluation, I appreciate the additional experiments and find them generally satisfactory. These experiments also help contextualize the limitations of TPS, which captures only pose-related discrepancies. For (W1.2), I acknowledge the misunderstanding on my part and appreciate the clarification that such hacking does not occur as easily as I initially assumed. For (W1.3), my concern stemmed from observations in [1,2] that pose signals from COLMAP can be unreliable in NVS tasks. This raised the question of whether VGGT might suffer from similar weaknesses. The supplementary robustness experiments convincingly address this issue, and I now find the use of deep networks like VGGT for pose-consistency evaluation conceptually appealing, similar to LPIPS for perceptual similarity. Overall, I appreciate the authors’ thoughtful responses.
> >
> > The only remaining concern (however, not affecting my rating) lies in W1.1. I would like to reiterate the fundamental reason why I raised W1.1. A key question is whether the proposed learning framework for truly transferable NVS comes with trade-offs—particularly in rendering quality—or what the nature of such trade-offs might be. Ideally, these questions would be examined through comparisons with prior work. However, because the paper challenges the validity of previous task formulations, it becomes difficult to perform fair comparisons in visual quality (as noted in the rebuttal: Rayzer cannot be evaluated under transferable NVS, while autoencoding settings do not correspond to true NVS). Consequently, although the method clearly excels in TPS, TPS alone may not fully capture potential degradation in rendering quality.
> >
> > 1. One motivation for this concern is the visible artifacts in the paper’s video results, particularly in scenes where Rayzer produces high-quality renderings (e.g., the first example on its homepage). This raises the possibility of a fundamental trade-off between camera-control accuracy and rendering fidelity. Conceptually, self-supervised NVS methods fall into three categories based on controllability:
> >     - (a) Transferable NVS (as proposed here), which controls cameras through a scene-to-scene transferable mapping;
> >     - (b) Autoencoding NVS (Rayzer, Less3Depend), which lacks controllability without a target view;
> >     - (c) SE(3)-controllable NVS, where Rayzer and Less3Depend map SE(3) poses into a latent space to enable camera control.
> >
> >     If the proposed approach improves control accuracy over (c) (which I believe it likely does) but significantly reduces rendering quality (which I am concerned about), then this would illustrate an inherent cost of achieving true NVS. Because the paper discusses only (a) and (b), omitting (c) may unintentionally overemphasize pose accuracy while leaving rendering quality underexamined.
> >
> > 2. That said, the paper does attempt a direct rendering-quality comparison with Rayzer under an autoencoding setting. Based on Table 4 alone, transferable NVS appears nearly “free”, achieving higher visual quality than Rayzer under the same conditions. I find this evidence sufficient, and although a joint comparison of controllability and visual quality against SE(3)-controllable NVS would be ideal, such an evaluation may fall outside the reasonable scope of a paper that introduces a new task definition and training paradigm.
> >
> > Taking all factors into account, I believe the paper is not perfect in some details, but its contributions are both significant and timely. Beyond revealing issues in current self-supervised NVS formulations, the proposed transferable training paradigm appears promising for broader applications. Currently, I believe this is a good paper (only waiting to read other reviewers' responses). If no more critical issues unexpectedly arise, I will update my rating to accept.
> >
> > Finally, I have a few questions for the authors:
> > - Q1. What motivated the choice of a 256-dimensional pose latent in RUST instead of the original 8-dimensional representation?
> > - Q2. Why does Rayzer’s SE(3) bottleneck hinder transferability? Is there an intuitive explanation?
> > - Q3. What was the rationale for using an L1+LPIPS loss rather than LVSM or Rayzer’s L2+Perceptual formulation?
> > - Q4. How stable is XFactor during training? (e.g., grad norm explosion)

---

> > > ### Author Response · Authors · 2025-11-27
> > > **Official Response by Authors [3/3]**
> > >
> > > ### Questions
> > > **Q1: 256-dimensional RUST pose latent**
> > > There exists no publicly available implementation of RUST and we view the paper’s principal contribution as predicting poses between full and partial views to help disentangle pixel information from the pose latents. In our reimplementation, we chose to use a 256-dimensional pose latent to most directly compare the full-to-partial masking with our own partial-to-partial masking scheme without the influence of a bottleneck. As we show in our ablations, adding an 8-dimensional bottleneck would likely improve the performance of RUST somewhat. However, the ablations also show that training a multi-view decoder end-to-end (without stereo-monocular pre-training) is likely the factor which most significantly degrades RUST’s performance.
> > >
> > > **Q2: Why does an SE(3) bottleneck hinder transferability**
> > > This is a great question, and one we have been thinking about ourselves. The RayZer pose encoder not only estimates the camera extrinsics (which are represented as elements of SE(3) ) for each view, but also a single set of camera intrinsics (which are modeled by a single focal length parameter) which are shared by all views in the sequence.  RayZer, in the original paper and in all of our experiments including ablations, is trained to reconstruct using context and target views which share the same intrinsics.  Thus, there is no impetus for the network to disentangle the predicted extrinsics and intrinsics into physically meaningful quantities.  At inference, we transfer only the camera extrinsics and not the intrinsics. Here, the model is seeing for the first time a combination of extrinsics extracted from two different cameras which are themselves not necessarily representations of the actual camera intrinsics but rather part of an entangled representation of camera and pose. Together, this likely contributes to the degradation relative to the unconstrained model.
> > > More broadly, we see this as illustrating how imposing multi-view geometric inductive biases can not only reduce performance but also create the illusion of interpretability: the model ostensibly produces camera intrinsics and extrinsics, yet those quantities need not correspond to true, disentangled geometry.
> > >
> > > **Q3: What was the rationale for using $L^1$ + LPIPS rather than $L^2$ + VGG Perceptual**
> > > It is known that existing image metrics still do not fully describe perceptual realism as observed by humans (For instance, see [1]). Thus, when choosing our loss, we went with what we found simply produced the best looking results — we experimented with training XFactor with both losses, and found that $L^1$ + LPIPS produced visually superior results in terms of our subjective judgment. Furthermore, in our experience, we have found that an $L^1$ loss consistently produces sharper, better looking images than an $L^2$ loss, despite many image metrics being aligned with the latter.  We also note that the choice of reconstruction loss is not fundamental to our approach and can be varied to suit the preferences of the user.
> > >
> > > **Q4: XFactor Training Stability**
> > > We found XFactor to be highly stable during training across a wide range of configurations and initialization conditions. Gradients consistently remained small without substantial deviation or spikes.  This was in contrast to RayZer, which in our experience we found to be highly sensitive to initialization conditions, and particularly those of the pose estimation head. (edited)

---

> ### Author Response · Authors · 2025-11-21
> **Camera Control via Sim-to-real Transfer**
>
> Dear Reviewer, we have prepared a proof-of-concept video demonstrating sim-to-real transfer as a means to control the camera pose in our webpage: https://xfactor-transferable-nvs.github.io
>
> We hope this helps with your review. Thank you!

---

> ### Author Response · Authors · 2025-11-27
> **Official Response by Authors [1/3]**
>
> We appreciate the reviewer’s thoughtful engagement and the intent to raise the rating. We are glad that the additional experiments resolved most of the initial concerns. We address the remaining concern and questions below:
>
> &nbsp;
> ### On the Trade-off between Rendering Quality and Control
> > The only remaining concern (however, not affecting my rating) lies in W1.1. I would like to reiterate the fundamental reason why I raised W1.1. A key question is whether the proposed learning framework for truly transferable NVS comes with trade-offs—particularly in rendering quality—or what the nature of such trade-offs might be.
>
> Your concern regarding the trade-off between pose fidelity and reconstruction quality is valid. We initially shared this concern, and we would like to explain how our experiments address it. To clarify this, we distinguish between the following two cases:
>
> **[Case 1] Hallucination, not a trade-off.**
>
> > ... One motivation for this concern is the visible artifacts in the paper’s video results, particularly in scenes where Rayzer produces high-quality renderings (e.g., the first example on its homepage) ...
>
> In many cases, RayZer appears to produce aesthetically better images simply because it ignores difficult target poses. For instance, in the video mentioned by the reviewer (the one with the cannon), RayZer ignores the target forward motion and instead hallucinates a lateral motion. Because this hallucinated view is closer to the training distribution, it looks sharper. Conversely, XFactor adheres to the difficult, out-of-distribution target trajectory, which naturally leads to more visual artifacts due to the difficulty of extrapolation. Therefore, we argue that this specific comparison should be viewed in terms of **hallucination**, rather than as a trade-off between reconstruction quality and pose fidelity.
>
> **[Case 2] The true trade-off.**
>
> > If the proposed approach improves control accuracy over (c) (which I believe it likely does) but significantly reduces rendering quality (which I am concerned about), then this would illustrate an inherent cost of achieving true NVS... Based on Table 4 alone, transferable NVS appears nearly “free”, achieving higher visual quality than Rayzer under the same conditions.
>
> To correctly assess this quality-fidelity trade-off without unfairly comparing against hallucinated results, we must limit our evaluation to in-distribution scenarios where all models are expected to maintain high pose fidelity. The critical question becomes: When we restrict the evaluation to the subset where the pose is correctly transferred, does XFactor suffer from quality degradation compared to RayZer?
>
> Our autoencoding experiment (Table 4) confirms that this is not the case. When we restrict the target poses to be strictly in-distribution, XFactor shows no quality degradation; in fact, it shows a "free" improvement over RayZer. This suggests there is no inherent quality tradeoff for achieving transferability.
>
> &nbsp;
> ### On the Difficulty of Joint Quality-Fidelity Comparison
>
> > That said, the paper does attempt a direct rendering-quality comparison with Rayzer under an autoencoding setting... I find this evidence sufficient, and although a joint comparison of controllability and visual quality against SE(3)-controllable NVS would be ideal, such an evaluation may fall outside the reasonable scope of a paper that introduces a new task definition and training paradigm.
>
> We appreciate the reviewer’s understanding regarding the challenges of jointly comparing controllability and visual quality. We note that for the transfer task, ground truth target views do not exist; thus, standard reference-based metrics like PSNR and LPIPS cannot be applied. We agree that the most rigorous quality assessment would ideally be performed directly on the transferred videos. However, this is a significant challenge that needs to be addressed in future research. This may even be an ill-posed problem, as inherent scale ambiguity makes it difficult to define a single "correct" ground truth for evaluation. Finally, we recall that we partially addressed this limitation by performing a distribution-level quality assessment using FID scores which does not require reference.

---

> ### Author Response · Authors · 2025-11-27
> **Official Response by Authors [2/3]**
>
> ### Clarification on the SE(3)-controllablility
> >   Conceptually, self-supervised NVS methods fall into three categories based on controllability:
> (a) Transferable NVS (as proposed here), which controls cameras through a scene-to-scene transferable mapping;
> (b) Autoencoding NVS (Rayzer, Less3Depend), which lacks controllability without a target view;
> (c) SE(3)-controllable NVS, where Rayzer and Less3Depend map SE(3) poses into a latent space to enable camera control.
>
> First, we argue that our method, XFactor, is "SE(3)-controllable." Our updated sim-to-real video shows that explicit SE(3) control is possible without manipulating the pose latent. The video can be found in our updated project page: https://xfactor-transferable-nvs.github.io/
>
> Also, we argue that RayZer and Less3Depend do not qualify as "SE(3)-controllable" for three fundamental reasons:
>
> 1. The learned representation is not algebraically SE(3). While these methods output 4x4 matrices, they do not preserve the group structure of SE(3). For instance, performing standard matrix inversion or multiplication on their learned representations does not correspond to the actual group inversion or composition of the rendered camera poses. Therefore, the representation is provably not SE(3); it is merely a 6-dimensional manifold that shares the topology of SE(3) but lacks its algebraic/geometric properties.
>
> 2. The exact same pose representation is decoded into different camera poses depending on the specific scene. Consequently, the user cannot predict which view the model will render: the same latent trajectory might produce motion towards the left in one scene, but towards the right in another. This behavior is consistent with our pose probe experiments, which suggest that the learned representation does not correspond to a unique target pose, but is instead entangled with the scene content.
>
> 3. We note that real-world transformations are more than SE(3). They include numerous non-pose factors, such as brightness changes, as also pointed out by Reviewer QYSv. We believe that attempting to force these complex transformations into a strict SE(3) bottleneck will inevitably result in a distorted and entangled representation. We avoid this bottleneck by training with a sufficiently high-dimensional representation; at inference time, the user can use a synthetic target video with pure SE(3) pose transformation that does not contain undesirable non-pose transformations.
>
>
> Lastly, we have updated our revision to include Less3Depend in our reference as it is highly relevant to our work.

---

> ### Comment · Reviewer_VrW1 · 2025-11-27
>
> Thanks for the dedicated response, and I really appreciate the authors' efforts in conducting these in-depth discussions.
>
> ---
>
> ### On the Trade-off between Rendering Quality and Control
>
> > (a) Transferable NVS (as proposed here), which controls cameras through a scene-to-scene transferable mapping;
> >
> > (b) Autoencoding NVS (Rayzer, Less3Depend), which lacks controllability without a target view;
> >
> > (c) SE(3)-controllable NVS, where Rayzer and Less3Depend map SE(3) poses into a latent space to enable camera control.
> >
> > ...we argue that this specific comparison should be viewed in terms of hallucination, rather than as a trade-off between reconstruction quality and pose fidelity.
>
> I highly agree with the use of the term "hallucination" when assessing (b). As a result, there is no trade-off between (b) and (a) (but an inability in camera control of (b)).
>
> ---
>
> ### Clarification on the SE(3)-controllablility
>
> > Also, we argue that RayZer and Less3Depend do not qualify as "SE(3)-controllable" for three fundamental reasons.
>
> Yes, I fully understand this and share the authors' view that, in their original forms, these methods belong to (b) autoencoding NVS. However, I notice there may have been a lack of clarity in my description of (c) (and I apologize for this misleading wording). I restate (c) as follows:
>
> - **(c) SE(3)-controllable NVS.** Less3Depend demonstrates that it is possible to employ a learnable linear layer to map an SE(3) pose into its latent pose space via fine-tuning, to enable camera control (arxiv [1]: Figure 9; Appendix B). I believe Rayzer could conduct a similar fine-tuning to achieve this.
>
> Consequently, I referred to these mapper-based variants as *SE(3)-controllable NVS* and was curious whether such forms of self-supervised methods could serve as comparable baselines. (That said, I now consider this point less critical: given that both methods suffer from gradient explosion, while the proposed XFactor is far more stable, delivering superior applicability in terms of training stability.)
>
> > [1] Wang, Haoru, et al. "The Less You Depend, The More You Learn: Synthesizing Novel Views from Sparse, Unposed Images without Any 3D Knowledge." arXiv preprint arXiv:2506.09885 (2025).
>
> ---
>
> ### Q1-Q4
>
> Thanks for the detailed explanation. I quite appreciate the superior training stability of XFactor. The stability advantage alone constitutes an important contribution to applicability, and it indicates that self-supervised NVS approaches may benefit from a regularized latent pose space compared to an ill-posed one from the perspective of training stability, which further demonstrates the value of this work.
>
> ---
>
> Again, thank you to the authors for their efforts in producing this solid work. In its current form, I believe this paper not only provides exceptionally valuable insights challenging conventional knowledge, but also establishes a solid, effective framework (covering both training and evaluation) for future self-supervised NVS research. Consequently, I believe it would better serve the community if featured at ICLR. Taking all these points into consideration, I am raising my rating to `strong accept, should be highlighted at the conference as spotlight or oral` to reflect its value in my view.

---

### Official Review · Reviewer_ZciQ · 2025-11-01

**Soundness:** 4
**Presentation:** 3
**Contribution:** 4
**Rating:** 8
**Confidence:** 4

**Summary:**

The paper argues that the correct criterion for self-supervised NVS is transferability: a pose representation extracted from one sequence should produce the same camera trajectory when used to render another scene. The authors show that recent pose-free/self-supervised methods (RayZer, RUST) learn interpolation latents that fail this test. The authors propose XFactor, a geometry-free model using pairwise pose estimation, stereo-monocular bootstrapping, and pose-preserving augmentations (e.g., inverse masks) to disentangle pose from content. The authors also propose a new metric named True Pose Similarity (TPS) quantifies transferability using oracle poses.

Experiments on datasets like RealEstate10K, DL3DV, MVImgNet, and CO3Dv2 demonstrate XFactor's superior transferability and pose probing accuracy, with ablations validating design choices.

**Strengths:**

The paper is insightful, which reframes NVS with the *transferability*. This is a fresh perspective, and the proposed TPS metric is a practical tool for evaluation the transferability of an NVS model.
Quality of this paper is strong, with large-scale experiments on diverse real-world datasets supporting claims of transferability; ablations thoughtfully dissect components like stereo-monocular training and augmentations. Paper presentation is quite clear, from problem analysis to method derivation, making complex ideas accessible.

**Weaknesses:**

1. The proposed TPS relies on an oracle (VGGT used in current experiments). VGGT is reasonable, but VGGT vs more conventional SfM (e.g., COLMAP) can differ—especially on difficult sequences (textureless regions, non-static content). The paper does not report robustness of TPS to oracle errors (noise, scale drift, missing frames), nor does it show if conclusions change with a different oracle.

2. The authors note blurring/warping for large baseline transfers and suggest this stems from determinism. This is an important limitation for practical use and should be explored more: does a generative renderer (diffusion model) fix artifacts? What are tradeoffs with photorealism vs pose fidelity?

3. The practical cost at inference (latency/memory) for rendering many target poses or long sequences is not reported.

**Questions:**

1. How might TPS change if the authors evaluated it with COLMAP alongside VGGT. How sensitive would TPS be to small perturbations (translation/rotation jitter) in the oracle poses?

2. Could a generative decoder plausibly reduce the blurring/warping seen at large baselines without degrading pose control — and if not, what trade-offs are expected between realism (e.g., FID/LPIPS) and pose fidelity (TPS)? Which probabilistic design choices seem most likely to preserve pose conditioning?

3. What are resource costs: expected latency and peak GPU memory for rendering one target view on common GPUs, and how latency and memory scale as the number of context views increases?

---

> ### Author Response · Authors · 2025-11-21
>
> We appreciate the reviewer's positive and constructive feedback, especially the recognition of our novel perspective and experimental rigor. We have addressed each of the concerns raised below.
>
> ### [Concern 1] TPS Oracle Comparison is Missing.
> > The proposed TPS relies on an oracle (VGGT used in current experiments). VGGT is reasonable, but VGGT vs more conventional SfM (e.g., COLMAP) can differ—especially on difficult sequences (textureless regions, non-static content). The paper does not report robustness of TPS to oracle errors (noise, scale drift, missing frames), nor does it show if conclusions change with a different oracle.
>
>
> We thank the reviewer for the suggestion to analyze the TPS metric in greater depth. We reflected this valuable feedback into our revision by conducting a comparative robustness analysis between VGGT and COLMAP in Appendix B. Our results demonstrate that COLMAP is fragile to minor input corruptions, often leading to large variance and excessively high failure rates. In contrast, VGGT demonstrates consistent robustness against these perturbations.
>
> &nbsp;
> ### [Concern 2]  Blurring/Warping Artifacts.
> > The authors note blurring/warping for large baseline transfers and suggest this stems from determinism. This is an important limitation for practical use and should be explored more: does a generative renderer (diffusion model) fix artifacts? What are tradeoffs with photorealism vs pose fidelity?
>
>
> It is a well-documented fact that deterministic renderers lead to blurry renderings when target areas are not covered by context views [1, 2]: For parts of the scene that are unseen in the context views, a deterministic model must predict the mean of all possibilities, which is invariably blurry. This is the source of our claim in the paper.
>
> However, we strongly agree with the reviewer that the interplay of a generative decoder with our self-supervised pose training objective is an exciting direction for future research, and in fact a research direction we are actively pursuing! Unfortunately, the answer to this question is still outstanding and will require a paper in its own right, so for now, we must defer this question to future work.
>
> We note that all of our claims are supported by comparisons to state-of-the-art deterministic self-supervised NVS models. To the best of our knowledge, no self-supervised *generative* models exist.
>
> We have added a discussion of this exciting future direction to our limitations section and conclusion. We would like to emphasize that this limitation is shared by deterministic NVS models supervised with ground-truth target poses, and therefore is not intrinsic to our self-supervised objective. We include this analysis in our response to reviewer QYSv which can be found in Figure 3 and Appendix B.
>
>
> [1] Tewari, Ayush, et al. "Diffusion with forward models: Solving stochastic inverse problems without direct supervision." Advances in Neural Information Processing Systems 36 (2023): 12349-12362.
>
> [2] Chan, Eric R., et al. "Generative novel view synthesis with 3d-aware diffusion models." Proceedings of the IEEE/CVF International Conference on Computer Vision. 2023.
>
>
> &nbsp;
> ### [Concern 3]  Computational Cost.
> > The practical cost at inference (latency/memory) for rendering many target poses or long sequences is not reported.
>
> Because our model is a fully feed-forward neural network, rendering is highly efficient. Our model renders novel views at up to 30 FPS on a single RTX A6000 GPU with the peak memory usage of less than 3 GB when using five context views (identical to our experimental setting). Please note that the RTX A6000 is less performant than more recent consumer-grade GPUs such as the RTX 5090.

---

> ### Author Response · Authors · 2025-11-21
> **Camera Control via Sim-to-real Transfer**
>
> Dear Reviewer, we have prepared a proof-of-concept video demonstrating sim-to-real transfer as a means to control the camera pose in our webpage: https://xfactor-transferable-nvs.github.io
>
> We hope this helps with your review. Thank you!

---

### Official Review · Reviewer_QYSv · 2025-11-01

**Soundness:** 3
**Presentation:** 4
**Contribution:** 3
**Rating:** 6
**Confidence:** 4

**Summary:**

This paper tackles self-supervised novel view synthesis by defining pose latent transferability as the key criterion for success. The proposed model, **XFactor**, achieves this through a geometry-free, self-supervised approach that combines pairwise pose estimation with curated data augmentations. By training on real-world videos without explicit geometric biases, XFactor learns transferable pose latents. To measure this capability, the authors introduce a new metric, True Pose Similarity (TPS). Experiments show that XFactor outperforms prior NVS transformers like RayZer and RUST on large-scale benchmarks. Furthermore, analysis confirms that its learned pose latents meaningfully correlate with actual camera poses, validating the approach.

**Strengths:**

1. This paper is well-written and has clear results/figures to present the results.  Datasets and baselines are well-described.

2. This work defines "transferability" as a discriminative property for novel view synthesis (NVS) and introduces a new metric (TPS) for its quantification. It argues that view reconstruction alone is insufficient for genuine NVS without this property.
3. The proposed model, XFactor, utilizes a pose-preserving, dual-masking data augmentation to disentangle the scene representation from the camera pose. This method achieves high transferability without relying on explicit geometric priors.
4. Empirical results show that XFactor substantially outperforms baselines across multiple datasets. The paper supports its claims with a comprehensive evaluation, including pose-probing to validate the learned latent space and detailed ablation studies that provide insights into the model's design.

**Weaknesses:**

- The novel views generated by XFactor, as seen on the project's homepage, suffer from noticeable blurring and distortion, even when the camera trajectory is accurately recovered. The paper would be stronger if it proposed a method to address these rendering imperfections. While the authors mention that blurring and warping occur for distant poses, a more in-depth analysis of these artifacts would add important nuance to the empirical results and better guide subsequent research.
- The paper's experimental validation would be more convincing if it included a comparison against other self-supervised learning objectives, such as those based on contrastive learning or mutual information. Additionally, the evaluation of the ViT-based architecture is incomplete without benchmarking it against more recent paradigms, including video diffusion models like ReCamMaster, which have shown strong performance on related tasks.
- The current experiments are limited to well-controlled datasets, leaving the performance of XFactor in more challenging, unconstrained real-world environments an open question. The paper does not provide insight into how the method would handle scenes with dynamic objects or significant photometric variations, such as changes in lighting or the presence of motion blur, which are common in real-world video.

**Questions:**

1. Are there plans for open-sourcing code and pretrained models to facilitate reproducibility？

2. Minor issue: The paper defers critical details of the training procedure to the appendix. For instance, the exact loss function is not defined in the main methodology. The description of the dual-masking augmentation is also incomplete, as it omits the probability of the no-masking case, making it unclear how frequently the transfer objective is applied during training.

---

> ### Author Response · Authors · 2025-11-21
> **Official Comment by Authors (1/2)**
>
> We appreciate the reviewer for the insightful and constructive feedback. We also thank the reviewer for acknowledging our novel transferability criterion and the strong empirical effectiveness of our proposed method. We have carefully considered all the points raised and have addressed every concern and question below.
>
> ### [Concern 1] Insufficient Analysis on Reconstruction Artifacts (Blurring / Distortion).
> > The novel views generated by XFactor, as seen on the project's homepage, suffer from noticeable blurring and distortion, even when the camera trajectory is accurately recovered. The paper would be stronger if it proposed a method to address these rendering imperfections. While the authors mention that blurring and warping occur for distant poses, a more in-depth analysis of these artifacts would add important nuance to the empirical results and better guide subsequent research.
>
> Indeed, our model's reconstruction is not perfect when rendering wide baseline, out-of-distribution, or out-of-sight target viewpoints. We thank the reviewer for carefully checking the experimental results and pointing out that we were missing in-depth discussion on this imperfect reconstruction when rendering these difficult target viewpoints.
>
>
> In particular, the reviewer has insightfully pointed out that the reconstruction often contains blurry or distorted artifacts for difficult viewpoints even though **the model has accurately recovered the camera pose.** As the camera poses are correctly inferred, these artifacts are most likely be coming from the **pure difficulty of NVS**, not intrinsic to our self-supervised objective. To confirm this, we have added a new analysis in Appendix B of our revision where we show that the **same problem happens for the supervised methods** (LVSM), illustrated in Fig 3. We hope this analysis will inform future work.
>
> &nbsp;
> ### [Concern 2] Lack of Comparison with Other SSL Methods.
> > The paper's experimental validation would be more convincing if it included a comparison against other self-supervised learning objectives, such as those based on contrastive learning or mutual information.
>
>
> We thank the reviewer for this suggestion, which we agree is an exciting research direction and indeed one that we have been pursuing. In our revision, we have added an experiment in the appendix where we train two versions of our pose estimation module without the transferability objective: In the first, we train with a contrastive SimCLR [1] objective and in the second we train with the mutual-information based VICReg [2] objective.  In both cases, we feed two augmentations to the pose encoder to produce pose embeddings and the losses are computed over the batch with respect to the objective. We do not train a renderer because we do not use the transferability objective.
>
> We report linear probe accuracy in Table 6 in the appendix and find that the SimCLR and VICReg objectives perform **far worse than our transferability objective** (and also all other design ablations in Table 3),  almost **completely failing to extract meaningful pose information**. This suggests that rendering may be a critical facilitator of the emergence of geometric reasoning. Despite these results, we believe our method can still be improved by incorporating ideas from previous work on self-supervised learning, and we are excited to pursue this in future work.
>
> [1] A Simple Framework For Contrastive Learning of Visual Representations (Chen at al., 2020)
>
> [2] VICReg: Variance-Invariance-Covariance Regularization for Self-Supervised Learning (Bardes et al. 2022)
>
> > Additionally, the evaluation of the ViT-based architecture is incomplete without benchmarking it against more recent paradigms, including video diffusion models like ReCamMaster, which have shown strong performance on related tasks.
>
> We agree that ReCamMaster is interesting related work and have added a discussion of this paper in Section 5.
>
> However, we note that ReCamMaster is a supervised model as it requires a dataset of ground-truth videos with camera poses. In contrast, our model is self-supervised and does not require any ground-truth pose annotations. Our model thus makes fundamentally different assumptions and has a different input-output behavior than ReCamMaster. We follow the standard NVS benchmark for view synthesis transformers established by RayZer and RUST.

---

> ### Author Response · Authors · 2025-11-21
> **Official Comment by Authors (2/2)**
>
> ### [Concern 3] Robustness in Unconstrained Environments.
> >The current experiments are limited to well-controlled datasets, leaving the performance of XFactor in more challenging, unconstrained real-world environments an open question. The paper does not provide insight into how the method would handle scenes with dynamic objects or significant photometric variations, such as changes in lighting or the presence of motion blur, which are common in real-world video.
>
>
> We thank the reviewer for this exciting suggestion. **In fact, this is exactly why we didn't model our camera representation to be SE(3).** Even in static scene datasets, there are numerous transformations that are not pose. Incorrectly modeling these subtle transformations altogether as an SE(3) transformation does not result in a good pose representation but a fragile, entangled and non-transferable representation. Instead, our representation serves as a superset of all these transformations. Indeed, our training objective can be leveraged to train models that transfer other scene properties. We are working on follow-up work where we have applied the same augmentation framework to relighting problem, rather than NVS.
>
> We also agree that an extension to dynamic novel view synthesis is an exciting area for future work, but this is beyond the scope of our present paper: we would not expect our current model to solve this problem zero-shot. Our core claim is that our method solves the true self-supervised novel view synthesis problem that was overlooked in the previous works. While the datasets we consider do indeed only consider static scenes and are thus not representative of the majority of real-world video, they are the standard benchmarks used in prior self-supervised NVS work. In these datasets, our method outperforms the previous SOTA RayZer, which just received an Oral at ICCV.
>
>
> &nbsp;
>
> ### [Questions]
> > Are there plans for open-sourcing code and pretrained models to facilitate reproducibility？
>
> Yes, we will publicly release all code and pre-trained model checkpoints at publication.
>
> > Minor issue: The paper defers critical details of the training procedure to the appendix. For instance, the exact loss function is not defined in the main methodology. The description of the dual-masking augmentation is also incomplete, as it omits the probability of the no-masking case, making it unclear how frequently the transfer objective is applied during training.
>
> Thank you for catching these omissions. The most relevant training details have been moved from the appendix to section 3.5. In our implementation, for each batch example, there exists a 5% chance that the image pair will not be masked.

---

> ### Author Response · Authors · 2025-11-21
> **Camera Control via Sim-to-real Transfer**
>
> Dear Reviewer, we have prepared a proof-of-concept video demonstrating sim-to-real transfer as a means to control the camera pose in our webpage: https://xfactor-transferable-nvs.github.io
>
> We hope this helps with your review. Thank you!

---

> ### Comment · Reviewer_QYSv · 2025-11-26
>
> I appreciate the detailed response, particularly the inclusion of the SimCLR/VICReg experiments in the appendix and the sim-to-real demonstration. I have read the other reviews and the authors' rebuttals.
>
> **[Concern 1] Insufficient Analysis on Reconstruction Artifacts (Blurring / Distortion) — Partially Addressed / Constructive Suggestions**
>
> I appreciate the authors' transparency in acknowledging that rendering imperfections for wide-baseline views are an inherent challenge in NVS, even for supervised methods like LVSM. The new analysis in Appendix B provides a fair context for these artifacts, and I agree that solving this completely is beyond the scope of this specific paper.
>
> However, to better bridge the gap between accurate trajectory recovery and visual fidelity,
>
> the author proposes integrating the current approach with a generative model. I would like to inquire about potential future directions for this work. For example:
>
>
>
> a) Could the features from the proposed model (XFactor)  be used to distill a diffusion model?
>
> b) Alternatively, could the generative model be used for refinement in a second stage?
>
> c) or some other promising solution for integrating current advantage with generative model?
>
>
> **SSL Baselines**: The addition of SimCLR and VICReg experiments in the appendix provides valuable insight. The poor performance of these contrastive/mutual-information objectives highlights the necessity of the proposed rendering-based transferability objective for geometric reasoning.
>
> **ReCamMaster Comparison**: I accept the authors' clarification. I agree that ReCamMaster, being a supervised method (requiring ground-truth pose) and based on a Diffusion Transformer (DiT) architecture, operates under a fundamentally different paradigm. Therefore, the current benchmarking against RayZer and RUST is sufficient and fair.
>
>
> **[Concern 3] Robustness & Minor Issues — Addressed**
>
> I acknowledge the authors' response regarding the scope of dynamic scenes and thank them for clarifying the training details (masking probability) and committing to open-sourcing the code.

---

> ### Author Response · Authors · 2025-11-27
>
> We thank the reviewer for highlighting these points and are also excited about the potential of our approach to combine with generative modeling.  In particular, your observations are consistent with several potential next steps we are currently exploring.
>
> > (a) Distillation into a generative model.
>
> Yes, learned XFactor features could potentially be used to distill or condition a diffusion model. The most straightforward application would be condition an image or video diffusion model on XFactor-extracted pose latents instead of ground truth camera poses, which could enable self-supervised conditioning.  Another interesting avenue we have yet to explore is whether the internal pixel tokens in both the pose encoder and renderer themselves constitute meaningful geometric features, as is the case in CroCo [1]. If so, they could potentially be used as conditioning inputs for geometry-aware diffusion models or as supervisory signals for distilling geometric priors into a generative backbone.
>
> [1] CroCo: Self-Supervised Pre-Training for 3D Vision Tasks by Cross-View Completion (Weinzaepfel et al. 2022)
>
> > (b) Second-stage refinement.
>
> A generative model could also serve as a refinement module applied after reconstruction, for instance, to abate blurring and distortion artifacts in out-of-distribution transfer and improve photorealism while preserving the geometry recovered accurately by our method.
>
> > (c) Extending XFactor into a generative framework.
>
> This is perhaps the most interesting direction in our opinion, and the one we are most excited about. We are currently experimenting with integrating diffusion sampling directly into the renderer in a similar style as SODA [2]. Our initial experiments suggest that naively incorporating diffusion into the renderer causes the synthesized images to drift from the correct trajectory, which does not occur in the deterministic case. Together this suggest potential innovation for a sampling/scheduling strategy which finds the right balance between determinism --- e.g. filling in information from available views where necessary to preserve geometry --- and stochasticity.
>
> [2] SODA: Bottleneck Diffusion Models for Representation Learning (Hudson et al. 2023)

---

> ### Comment · Reviewer_QYSv · 2025-11-27
>
> I also believe that extending XFactor into a generative framework is a promising direction for achieving both precise camera trajectories and high-quality scene generation.
>
> The authors have effectively addressed my questions, and I have no further concerns. Consequently, I will maintain my current positive score, given the novelty and potential impact of this work on the field. However, I recommend that the authors incorporate the discussions from the review process (including other reviewers' discussions) into the revised manuscript.

---

> > ### Author Response · Authors · 2025-11-27
> >
> > We sincerely thank the reviewer for the engaging discussion and for confirming that the additional experiments (SimCLR/VICReg baselines) and demonstrations (Sim-to-Real) have effectively addressed your concerns. We are glad that we could reach a consensus regarding the scope of the rendering artifacts and the potential of our method for future generative approaches.
> >
> > We noticed that while you stated you have “no further concerns” and that the baselines provided “valuable insight,” the score remains at a 6 (marginal accept). As the rebuttal period comes to a close, we respectfully ask if you might consider raising your score to reflect the improvements made to the manuscript during this discussion. Specifically, since the original evaluation, the paper now includes:
> >
> >  - **Quantitative SSL Baselines**: Proving the necessity of our transferability objective over contrastive/mutual-information methods.
> >
> > - **Real-World Validation**: Investigating robustness via the Sim-to-Real transfer demo.
> >
> >  - **Clarified Scope**: Contextualizing the rendering artifacts as a shared challenge with supervised SOTA methods.
> >
> > Given that these additions directly addressed the weaknesses cited in the initial review, and considering the strong support from the other reviewers, we hope you might consider adjusting the rating to align with the improved state of the submission.
> >
> > Regardless of the outcome, we sincerely appreciate the time you took to help us strengthen this work.

---

### Author Response · Authors · 2025-12-01
**Note to AC: Summary of Rebuttal Discussions (1/3)**

Dear AC — thank you for handling this difficult situation. Below is a summary that (i) lists our paper's contributions, (ii) reviewer assessments, (iii) summarizes how each reviewer's concerns were addressed in the rebuttal/revision, and (iv) records the concrete reviewer updates visible in the OpenReview thread. All items below reflect the current OpenReview thread for Submission **13980**.

* * *

Contributions
-------------

*   **Novel Problem Formulation:** We introduce **transferability** as the correct operational criterion for _true_ self-supervised novel-view synthesis (NVS): a pose latent extracted from one video should reproduce the same camera trajectory when re-used in another scene.

*   **Methodological Novelty:** We propose **XFactor**, a geometry-free self-supervised model that combines stereo-monocular bootstrapping with pose-preserving data augmentations to disentangle camera pose from scene content and enable transfer.

*   **New Evaluation Metric:** We introduce **TPS (True Pose Similarity)**, an oracle-based metric that quantitatively measures transferability and exposes failure modes of prior pose-free/self-supervised NVS methods.

*   **Empirical Significance:** Extensive empirical validation across multiple datasets (RealEstate10K, DL3DV, MVImgNet, CO3Dv2) shows XFactor substantially improves TPS and pose-probe accuracy vs. prior pose-free methods; we also include ablations and a sim→real camera control demonstration.


* * *

Positive assessments
--------------------------------------------------------------------------------
All four reviewers recognized the novelty, clear motivations, technical soundness, and empirical contributions of the work; several reviewers used strong positive language above to indicate their confidence in the paper's contributions.

*   **Reviewer 9vfk:** called our reframing of NVS in terms of transferability "**insightful**" and later wrote the rebuttal "**successfully addressed all my initial concerns**,"  that our central contribution is "**novel and impactful**", and that our paper is "**exceptionally well-written**". Upon recieving our rebbutal, they stated their intent to update their rating from a **4 to an 8** and explicitly voted to accept.

*   **Reviewer VrW1:** Stated our main idea is "**insightful and impactful**"" and finds out "**very strong**". After a discussion during the rebuttal, they stated the work "**not only provides exceptionally valuable insights challenging conventional knowledge, but also establishes a solid, effective framework (covering both training and evaluation) for future self-supervised NVS research**" and would "**better serve the community if featured at ICLR**". They explicitly stated their intent to raise their score from a "**6 to a 10**".

*   **Reviewer QYSv:** described the extra experiments (SimCLR/VICReg baselines and sim→real demo) as providing "**valuable insight**". While we did not finish the discussion with the reviwer, they noted the "**novelty and potential impact**" of our work, stated intent to "**maintain [their] current positive score**".

*   **Reviewer ZciQ:** provided a long, positive assessment, explicitly praising the transferability framing and TPS as a "**fresh perspective**" and "*insightful*". The reviewer noted that the "**quality of this paper is strong**", the ablations were "**thoughtful**", and that "**paper presentation is quite clear, from problem analysis to method derivation, making complex ideas accessable**". The reviewer did not respond to our rebuttal before the discussion was frozen.

At the time the discussion was frozen, our paper had the following scores:

* 9vfk: 8
* VrW1: 10
* QYSv: 6 (Had not finished rebuttal discussion)
* ZciQ: 8 (Had not yet responded to our rebuttal)

---

> ### Author Response · Authors · 2025-12-01
> **Note to AC: Summary of Rebuttal Discussions (2/3)**
>
> Addressing Reviewer Concerns in the Rebuttal
> ----------------------------------------------------------------
>
>
> ### **[Reviewer 9vfk]**
>
> **Concerns raised:**
> - Argued the paper's "transferability" phrasing risked over-claiming / wanted the paper's scope.
> - Asked for evidence on **robustness** and stronger ablations.
>   (See reviewer text noting the scope concern and later comment praising the detailed rebuttal.)
>
> **How we addressed them:**
> - Clarified scope in the paper and rebuttal (made clear that existing **supervised feed-forward and single-scene NVS methods are automatically transferable** and our claims concern only self-supervised NVS).
> - Added additional robustness experiments in which assumptions behind our proposed transferability objective are progressively corrupted, showing that our objective **remains robust under significant perturbations**.
> - Posted sim→real demo on the project page to show an example of how practical camera control can be achieved.
>
>
> * * *
>
> ### **[Reviewer VrW1]**
>
> **Concerns raised:**
> - Questioned whether **TPS** (oracle-based) fully captures **perceptual/rendering quality** and whether transferability trades off rendering fidelity; requested additional perceptual evaluations.
> - Concerns about **oracle fragility** (COLMAP vs VGGT) and whether VGGT might be unreliable for TPS.
> - Asked about training stability / trade-offs in comparisons to prior work.
>
> **How we addressed them:**
> - Added perceptual analyses (perceptual metrics for inference-time masking reported in Appendix Table 5) confirming that our method **achieves exceptional rendering quality when transferred poses are near the distribution of target poses**.
> - Ran oracle robustness experiments (VGGT vs COLMAP, noise/perturbation studies) and reported results in the appendix — showing VGGT is sufficiently stable in our regime and that TPS remains meaningful under realistic perturbations.
> - Clarified training stability benefits of XFactor vs RayZer (noting gradients / initialization sensitivity differences) in replies.
>
>
> * * *
>
> ### **[Reviewer QYSv]**
>
> **Concerns raised:**
> - Highlighted **blurring / distortion** artifacts for wide-baseline / out-of-distribution rendered views and wanted more analysis and discussion of remedies (e.g., generative refinements).
> - Requested clarity on training details (e.g., dual-masking probabilities) and suggested comparisons to SSL baselines.
>
> **How we addressed them:**
> - Added an appendix analysis contextualizing artifacts showing supervised/deterministic methods also struggle and that such artifacts are **not due to our method, but rather an inherent property of deterministic NVS models** .
> - Added ablations using SimCLR / VICReg SSL objectives which show that contrastive/mutual-information objectives do **not** extract similarly usable pose latents.
> - Moved key training details (masking probabilities, loss specifics) into the main text and clarified protocol choices in the rebuttal.
> - Provided sim→real demo and discussed future generative refinement directions (distillation / second-stage refinement / diffusion conditioning).
>
>
> * * *
>
> ### **[Reviewer ZciQ]**
>
> **Concerns raised:**
> - Questioned robustness of TPS using different oracles: VGGT vs COLMAP (similar to VrW1).
> - Requested clarification on blurring and warping artifacts for out-of-distribution transfer and how this might be abated with a generative model (similar to QYSv)
>
> **How we addressed them:**
> - Ran targeted oracle-robustness experiments directly comparing VGGT and COLMAP (including controlled noise and perturbation studies) and reported these results in the appendix; these analyses show **VGGT provides stable pose estimates in our evaluation regime** and that **TPS remains a meaningful aggregate metric under significant distortions of the input frames**.
>
> - Added a focused appendix analysis that quantifies blurring/warping artifacts for out-of-distribution transfers and compares these artifacts to supervised and deterministic NVS baselines, demonstrating that such artifacts **are a common limitation of deterministic renderers rather than a failure unique to our model**.
>
> - Expanded the manuscript and rebuttal with a concrete discussion of the potential integration of our approach with a generative model.

---

> ### Author Response · Authors · 2025-12-01
> **Note to AC: Summary of Rebuttal Discussions (3/3)**
>
> Summary and Final Remarks
> ----------------------------------------------------------------
> There is a unanimous consensus among all four reviewers regarding the novelty and significance of our "transferability" perspective. During the rebuttal phase, we successfully addressed all the concerns through extensive additional experiments and clarifications. In particular, Reviewers VrW1 and 9vfk explicitly acknowledged these improvements and raised their scores significantly, from 6 -> 10 and 4 -> 8, respectively, for final scores of 10, 8, 8, 6.
>
> We emphasize that this work offers a non-incremental, paradigm-shifting perspective on the fundamental problem of self-supervised NVS. We reveal that existing self-supervised NVS baselines are not controllable, and are currently evaluated on objectives that do not take this controllability into account. We provide transferability as an effective surrogate objective to measure controllability via the intuitive "same representation, same pose" principle. Using this criterion, we demonstrate that the direct latent space manipulation relied upon by prior methods is neither necessary nor sufficient. Regarding insufficiency, existing methods cannot achieve true controllability through direct latent manipulation because they lack transferability. Regarding necessity, transferability alone is already sufficient to achieve control without manipulating the latents, which we demonstrate through our sim-to-real example.
>
> We believe our work timely establishes transferability as the correct operational foundation for self-supervised NVS that was previously missing. We hope this insight reorients the community away from overfitting to non-transferable criteria and focuses future research on the right path to achieving true controllability.

---

### Meta-Review · Area_Chair_wz7m · 2026-01-06

**Summary:**

Long discussions between authors and reviewers. Most of the concerns are resolved. Two reviewers explicitly mentioned that they are impressed with the response with score raise. In AC's opinion, remaining concerns are minor or beyond the scope of the paper.

**Reviewer Concerns:**

Remaining Concerns:

1. **Missing comparisons to SSL baseline methods and Video-based models**. (QYSv) The paper provides some contrastive learning baselines. However, these methods are not adaptive and stick to the original frameworks. AC thought such adaptation is beyond the scope of the paper and thus this concern is minor.

2. **Generalization to dynamic scenes**. (QYSv) I agree that this is a valid extension to the paper, and is beyond the scope of this paper.

Resolved Concerns:
1. **Low Rendering Quality** (QYSv, ZciQ, VrW1, 9vfk). Added results in rebuttal to show that this issue also exists in baseline models.

2. **Generalization to complex real scenes** (QYSv, 9vfk) Video results are provided during rebuttal.

3. **Intrinsic error of using VGGT as evaluation metric** (ZciQ, VrW1). Provided additional results in rebuttal.

4. **Lack of perceptual evaluation** (VrW1). Dense discussion with the reviewer.

5. **Controllability of the Latent Camera** (9vfk) This is a good point. The author said that the control can be completed by first rendering with desired pose and then transfer. Some proof-of-concept demos are provided during rebuttal. AC feels that this is a temporal solution but seems to be satisfied to the reviewer.

6. **The method might optimize towards the new metric proposed by the authors** (9vfk) AC thought that this is a good point though hard to be evaluate. Reviewer is satisfied with the response.

**Reviewer Scores:**

6 8 8 10 from 6 4 8 6


VrW1: The original rating is 6. VrW1 explicitly mentioned that he would change to 10.
9vfk: The original rating is 4. 9vfk explicitly mentioned that he would change to 8.

---

### Decision · Program_Chairs · 2026-01-26

Accept (Oral)